# Stratified GRPO: Handling Structural Heterogeneity in Reinforcement Learning of LLM Search Agents

**Mingkang Zhu** [1]   **Xi Chen** [2]   **Bei Yu** [1]   **Hengshuang Zhao** [2]   **Jiaya Jia** [3]

## Abstract

Large language model (LLM) agents increasingly rely on external tools such as search engines to solve complex, multi-step problems, yet their rollouts are structurally heterogeneous: variations in tool-call number, placement, and outcomes induce distinct behaviors and reward distributions. As a result, policy gradient methods with a single global baseline suffer from *cross-stratum bias*, an "apples-to-oranges" comparison that distorts credit assignment and impedes exploration. To address this issue, we propose *Stratified GRPO*. Its core component, *Stratified Advantage Normalization* (SAN), partitions trajectories into homogeneous strata based on structural properties and computes advantages locally within each stratum, ensuring comparisons only among true peers. We show that SAN eliminates cross-stratum bias, yields conditionally unbiased unit-variance estimates within strata, and preserves the global unbiasedness and unit-variance properties of standard normalization, resulting in a more reliable learning signal. To improve robustness in finite-sample regimes, we further linearly blend SAN with the global estimator. Experiments on factual QA and deep-research agent benchmarks demonstrate that Stratified GRPO consistently outperforms GRPO by up to 12.6 points, achieving higher training rewards, improved training stability, and more effective search policies. These results establish structure-aware advantage normalization as an effective correction for RL of search agents with structurally heterogeneous rollouts.

[1]The Chinese University of Hong Kong [2]The University of Hong Kong [3]The Hong Kong University of Science and Technology. Correspondence to: Mingkang Zhu <mkzhu23@cse.cuhk.edu.hk>, Jiaya Jia <jia@cse.ust.hk>.

*Proceedings of the $43^{rd}$ International Conference on Machine Learning*, Seoul, South Korea. PMLR 306, 2026. Copyright 2026 by the author(s).

## 1. Introduction

Large Language Models (LLMs) (Achiam et al., 2023; DeepSeek-AI et al., 2025; Team, 2024) are increasingly augmented with external tools such as search engines, creating agents to tackle complex tasks (Schick et al., 2023; Yao et al., 2023; Jin et al., 2025; 2024; Trivedi et al., 2023; Asai et al., 2023). Reinforcement learning (RL) has emerged as a powerful paradigm for training these agents, allowing them to learn sophisticated multi-turn reasoning and tool-use strategies directly from outcome rewards (Chen et al., 2025; Jin et al., 2025; Song et al., 2025; Zheng et al., 2025).

A key, and often overlooked challenge for applying standard RL to search agents lies in the structural heterogeneity of their trajectories. Unlike conducting RL for conventional LLMs, where trajectories are sampled from the policy model and follow a relatively homogeneous pattern, the trajectories of search agents differ markedly in their structure due to the number, placement, and outcomes of search invocations. For instance, a trajectory with zero search calls relies solely on parametric memory and is qualitatively different from one that synthesizes information from multiple external documents, as the retrieved information can substantially alter subsequent generations and induce distinct behavior modes and, consequently, different reward distributions. Standard policy gradient methods compute advantages using a single global baseline, implicitly assuming all trajectories are comparable. This creates a flawed "apples-to-oranges" comparison for search agents. In this work, we identify and formalize this fundamental issue as **cross-stratum bias**. As we show, this structural flaw distorts credit assignment and hinders exploration of complex multi-step search strategies, leading to suboptimal policies.

We propose **Stratified GRPO** to address this fundamental issue. Instead of using a single global baseline, Stratified GRPO uses Stratified Advantage Normalization (SAN), a principled advantage estimator designed for heterogeneous action spaces. The core idea is simple yet powerful: partition trajectories into homogeneous strata, based on their structural properties (i.e., the number of search calls), and then compute advantages within each stratum. By construction, SAN is free from cross-stratum bias, ensuring that each trajectory is evaluated only against its true peers. Our

theoretical analysis formalizes the benefits of this approach. We show that SAN removes the cross-stratum bias inherent in global baselines, ensuring fair credit assignment. We further prove that SAN is conditionally unbiased and has unit variance within each stratum, acting as a pure and scale-stable learning signal. Critically, we demonstrate that SAN achieves these superior conditional properties while matching the global unbiasedness and unit variance of standard normalization methods. To enhance practical stability in finite-sample regimes, we further introduce Blended Advantage that robustly combines SAN with the global estimator.

We validate Stratified GRPO through comprehensive experiments on diverse factual QA and deep-research agent benchmarks. The results demonstrate the clear superiority of our method. Stratified GRPO consistently outperforms standard GRPO, achieving an improvement of up to 12.6 points. Furthermore, Stratified GRPO also exhibits higher training rewards, greater training stability, and learns more effective search policies than standard GRPO. These findings provide empirical evidence that our approach mitigates cross-stratum bias in the evaluated search-agent settings. Our main contributions are as follows:

- We identify and formalize **cross-stratum bias**, a fundamental challenge in policy gradient methods for LLM search agents. We provide a theoretical decomposition that proves that this bias arises from using a global baseline across structurally heterogeneous trajectories.

- We propose **Stratified GRPO**, a principled RL algorithm that eliminates this cross-stratum bias. Its core component, Stratified Advantage Normalization (SAN), partitions trajectories into homogeneous strata and computes advantages locally, ensuring a fair and stable credit assignment.

- We provide a rigorous theoretical analysis of SAN, proving that it eliminates cross-stratum bias and is conditionally unbiased and has unit variance within each stratum. Crucially, SAN achieves these superior conditional properties while preserving the global unbiasedness and unit variance of standard normalization, yielding a more pure and scale-stable learning signal.

- We demonstrate through extensive experiments on diverse factual QA and deep-research agent benchmarks that Stratified GRPO substantially outperforms GRPO by up to 12.6 points. Our method achieves higher training rewards, improves training stability, and learns more effective search policies.

## 2. Related Work

**Reinforcement Learning for Large Language Models.** Reinforcement learning (RL) (Kaelbling et al., 1996; Sut-

ton & Barto, 2018) has become a central component in post-training large language models (LLMs). The most widely adopted paradigm is Reinforcement Learning from Human Feedback (RLHF), which learns a reward model from human preferences and then optimizes a policy with RL algorithms, such as Proximal Policy Optimization (PPO) (Ouyang et al., 2022; Schulman et al., 2017). Although effective, RLHF can be computationally expensive and brittle due to reward model training and distribution shift. To reduce the cost and instability of explicit reward modeling, direct alignment methods like DPO optimize preference data without training a separate reward model (Rafailov et al., 2023; Zhu et al., 2025b;c; Meng et al., 2024). A complementary line of work targets reasoning by exploiting verifiable outcomes using Reinforcement Learning with Verifiable Rewards (RLVR) (DeepSeek-AI et al., 2025; Shao et al., 2024; Ahmadian et al., 2024; Yu et al., 2025; Chen et al., 2026). A prominent approach is Group Relative Policy Optimization (GRPO) (Shao et al., 2024), which removes PPO's dependency on learning a value function by using group-based baselines. RLOO (Ahmadian et al., 2024) revisits REINFORCE (Williams, 1992) with simplifications tailored to LLM training. However, most of these advances target general conversational capabilities. By contrast, the systematic study of RL algorithms for LLM agents, especially search agents that interleave generation with search over external information and require long-horizon reasoning, remains underexplored. Our work addresses this gap by providing a systematic analysis of the LLM search agent setting and proposing RL algorithms tailored to its unique challenges.

**Large Language Model Search Agents.** Large language models (LLMs) (Achiam et al., 2023; Team, 2024; Zhao et al., 2023; Yang et al., 2024; Zhu et al., 2025a; Yang et al., 2025; Dubey et al., 2024; DeepSeek-AI et al., 2025) exhibit strong reasoning capabilities (DeepSeek-AI et al., 2025). Building on this capacity, recent work has developed agentic workflows that equip LLMs with external tools for complex problem solving (Schick et al., 2023). A prominent instantiation is the LLM search agent, which treats a search engine as a callable tool at inference time (Schick et al., 2023): the model iteratively proposes queries, retrieves evidence, and updates its reasoning based on retrieved documents (Trivedi et al., 2023). Two main approaches have emerged. One line of work uses carefully designed prompts to instruct LLMs to interleave reasoning and retrieval (Trivedi et al., 2023; Yao et al., 2023). Another line curates trajectories that mix reasoning with search and then applies supervised fine-tuning (Schick et al., 2023; Asai et al., 2023). More recently, several studies (Chen et al., 2025; Jin et al., 2025; Song et al., 2025; Zheng et al., 2025) show that complex search-and-reasoning behaviors can be acquired directly from outcome-based rewards using RL algorithms such as PPO or GRPO. However, these RL applications typi-

cally adopt general-purpose algorithms without addressing the specific intricacies of the search agent setting. Our work identifies and formalizes cross-stratum bias as a fundamental challenge for LLM search agents, arising from heterogeneous sampled trajectories that use different search strategies. To overcome this, we propose Stratified GRPO, a principled algorithm designed to eliminate this bias and improve learning more effective search policies for LLM search agents.

## 3. Methods

In reinforcement learning (RL) for multi-turn search agents, an agent might choose different search strategies (e.g., varying search counts, timing, and queries), thus trajectories sampled from the same policy can rely on different amount of external information, belong to distinct structural groups, and are not directly comparable. We show that the structural heterogeneity of trajectories induces a **cross-stratum bias** whenever advantages are computed with baselines that ignore the heterogeneity driver. We then introduce **Stratified Advantage Normalization (SAN)**, an estimator that partitions trajectories into homogeneous strata and normalizes advantages therein, and analyze its statistical and structural properties.

### 3.1. RL for Multi-Turn Search Agents

Following Jin et al. (2025), we formulate multi-turn search agent training as an RL problem. The agent, parameterized by a policy $\pi_\theta$, interacts with a search engine by interleaving token generation with search queries. For a given prompt $x \sim \mathcal{D}$, the agent generates a trajectory $\tau \sim \pi_\theta(\cdot \mid x)$, which consists of a sequence of actions: generating tokens and issuing searches. Upon completion, the trajectory receives a scalar reward $R(\tau)$ that reflects the quality of the final response. The objective is to maximize the expected reward:

$$\max_\theta J(\theta) = \mathbb{E}_{x\sim\mathcal{D}, \tau\sim\pi_\theta(\tau|x)}[R(\tau)], \qquad (1)$$

which is usually optimized using policy gradient methods.

### 3.2. Cross-Stratum Bias in Policy-Gradient Baselines

**Intuition.** In search agents, two trajectories sampled from the same policy may not be comparable: e.g., a no-search rollout relies purely on parametric memory, whereas a multi-search rollout conditions on retrieved evidence and may explore systematically different reasoning paths and have different reward distributions. When we compute advantages using a single global baseline, trajectories from different structural regimes are compared against one another, introducing a **cross-stratum bias** in the advantage. This bias distorts credit assignment and can suppress exploration of

complex search behaviors. We formalize the cross-stratum bias by decomposing the global advantage below:

**Notation.** Let $B = \{\tau_1, \ldots, \tau_K\}$ denote a batch of $K$ trajectories sampled i.i.d. from $\pi_\theta$ for a fixed prompt $x$. The batch is partitioned into $I$ non-empty strata $B_0, \ldots, B_{I-1}$ according to a predefined structure (i.e., search count), with $|B_k| = n_k$. Each trajectory $\tau_i$ has reward $R_i$. Let $\bar{R}_{\text{global}} = \frac{1}{K}\sum_{j=1}^{K} R(\tau_j)$ be the global mean reward of the batch, and define the stratum-specific mean $\widehat{\mu}_k = \frac{1}{n_k}\sum_{\tau_i \in B_k} R_i$. This gives rise to two natural advantage estimators:

$$\hat{A}_G(\tau_i) = R_i - \bar{R}_{\text{global}} \quad \text{(global)},$$

$$\hat{A}_S(\tau_i) = R_i - \widehat{\mu}_k \quad \text{(stratified, } \tau_i \in B_k\text{)}.$$

**Advantage Bias.** The following result shows how the global advantage differs systematically from the stratified advantage.

**Proposition 3.1** (Advantage Decomposition). *For any trajectory $\tau_i \in B_k$, the global advantage decomposes as*

$$\hat{A}_G(\tau_i) = \hat{A}_S(\tau_i) + \underbrace{(\widehat{\mu}_k - \bar{R}_{\text{global}})}_{\textit{cross-stratum bias}}.$$

*(Proof follows directly from definitions).* This decomposition highlights a structural flaw: the **cross-stratum bias** is a deterministic stratum-specific offset applied to every trajectory in that stratum, unfairly penalizing trajectories from low-reward strata (e.g., those attempting newly discovered complex search strategies may initially fail), thereby suppressing exploration of diverse search strategies.

### 3.3. Stratified Advantage Normalization: Definition and Theoretical Guarantees

To eliminate the cross-stratum bias, we propose **Stratified Advantage Normalization (SAN)**. By partitioning the batch into homogeneous strata and computing advantages *locally*, SAN ensures that each trajectory is evaluated solely against its structural peers, yielding a conditionally unbiased and scale-stable signal.

**Definition 3.2.** For a given prompt $x$, partition the batch of trajectories into strata $\{B_k(x)\}$ based on a chosen partitioning function (i.e., the search count for search agents). The SAN advantage for a trajectory $\tau_i \in B_k(x)$ is defined as

$$A_{\text{SAN}}(\tau_i) = \frac{R(\tau_i) - \widehat{\mu}_k(x)}{\widehat{\sigma}_k(x) + \varepsilon}, \qquad (2)$$

where $\widehat{\mu}_k(x)$ and $\widehat{\sigma}_k(x)$ are the empirical mean and standard deviation of rewards in stratum $B_k(x)$, $\varepsilon > 0$ is a small constant for numerical stability.

$A_{\text{SAN}}$ inherits the desirable scale invariance of standard normalization, making it robust to the scale and shift of rewards, a desirable property formalized in Appendix A.1.

**Comparison to Global Normalization.** To appreciate the design of SAN, we contrast it with the standard Global Normalization (GN) used in GRPO (Shao et al., 2024). GN standardizes rewards globally, i.e.,

$$A_{\text{GN}}(\tau_i) = \frac{R(\tau_i) - \bar{R}_{\text{global}}}{\hat{\sigma}_{\text{global}} + \varepsilon},$$

where $\hat{\sigma}_{\text{global}}$ is the global standard deviation of rewards. $A_{\text{GN}}(\tau_i)$ ignores the structural heterogeneity of the search trajectories. This can be made precise by algebraically expressing the GN advantage in terms of the SAN advantage.

**Proposition 3.3** (Exact Advantage Decomposition). *For any fixed batch partition $\{B_k(x)\}_{k=0}^{I-1}$ and any $\tau_i \in B_k(x)$,*

$$A_{\text{GN}}(\tau_i) = \underbrace{\frac{\hat{\sigma}_k(x) + \varepsilon}{\hat{\sigma}_{\text{global}}(x) + \varepsilon}}_{:=\alpha_k(x)} A_{\text{SAN}}(\tau_i) + \underbrace{\frac{\hat{\mu}_k(x) - \bar{R}_{\text{global}}(x)}{\hat{\sigma}_{\text{global}}(x) + \varepsilon}}_{:=\Delta_k(x)}. \quad (3)$$

*(Proof in Appendix A.2).* The decomposition in Proposition 3.3 reveals that the GN advantage equals a rescaled SAN advantage plus a **cross-stratum offset** $\Delta_k$, which is the source of its systematic bias.

The decomposition in Equation (3) carries over directly to the gradient estimators. The GN gradient, $\hat{g}_{\text{GN}}$, decomposes into a SAN-like term and an additional bias-inducing term:

$$\hat{g}_{\text{GN}}(x) = \frac{1}{K} \sum_i A_{\text{GN}}(\tau_i) \nabla_\theta \log \pi_\theta(\tau_i \mid x)$$

$$= \frac{1}{K} \sum_k \sum_{\tau_i \in B_k} \alpha_k A_{\text{SAN}}(\tau_i) \nabla_\theta \log \pi_\theta(\tau_i \mid x) +$$

$$\underbrace{\frac{1}{K} \sum_k \sum_{\tau_i \in B_k} \Delta_k \nabla_\theta \log \pi_\theta(\tau_i \mid x)}_{\text{Bias from Cross-Stratum Offset}}. \quad (4)$$

The decomposition in Equation (4) reveals a structural flaw in the GN gradient, driven by the cross-stratum offset $\Delta_k$. This term couples reward differences across strata with the policy's score vectors $\nabla_\theta \log \pi_\theta(\tau_i \mid x)$, introducing a systematic bias that persists whenever strata are heterogeneous. This fundamentally distorts the learning signal by forcing local credit assignment to depend on global statistics.

**Cross-Stratum Bias Hurts Exploration.** The decomposition in Equation (3) makes the failure mode concrete. The cross-stratum offset $\Delta_k$ has the same sign as the difference between the stratum mean and the global mean $(\hat{\mu}_k(x) - \bar{R}_{global})$. When the policy begins to discover complex multi-search strategies, such trajectories often receive lower average rewards than shorter, low-risk trajectories the model has already mastered, due to retrieval errors and long-horizon compounding mistakes. This implies $\hat{\mu}_k < \bar{R}_{global}$.

Consequently, $\Delta_k$ becomes negative, effectively imposing a penalty that suppresses the gradients for these strata. This systematic suppression forces the policy to abandon complex search behaviors in favor of short, low-risk strata, making it hard for the policy to explore and improve complex search strategies that are potentially superior. This behavior is consistent with our empirical observation in Figure 1.

**Analysis of Advantage Signal Quality.** Under trajectory heterogeneity, an ideal advantage signal should be unbiased and have a consistent scale within each stratum, so that different structural regimes receive comparable learning dynamics. The following theorem formalizes why SAN provides a more reliable learning signal.

**Theorem 3.4** (Conditional Properties of SAN and GN). *Let $\varepsilon = 0$. Let $\mu_k(x), \sigma_k^2(x) \neq 0$ be the population reward mean and variance of each stratum $k$, and $\mu(x), \sigma^2(x) \neq 0$ be the global statistics. In the large-sample limit, the conditional properties for any stratum $k$ are:*

*(a) Conditional Expectation:*

$$\mathbb{E}[A_{\text{SAN}} \mid k, x] = 0, \quad \mathbb{E}[A_{\text{GN}} \mid k, x] = \frac{\mu_k(x) - \mu(x)}{\sigma(x)}.$$

*(b) Conditional Variance:*

$$\text{Var}(A_{\text{SAN}} \mid k, x) = 1, \quad \text{Var}(A_{\text{GN}} \mid k, x) = \frac{\sigma_k^2(x)}{\sigma^2(x)}.$$

*(Proof in Appendix A.3).* Theorem 3.4 shows that SAN provides a *pure* (zero-mean) and *scale-stable* (unit-variance) learning signal within every stratum. In contrast, GN introduces a systematic bias proportional to the cross-stratum mean difference and inversely proportional to a variance that fluctuates based on reward heterogeneity. By eliminating this structural bias, SAN ensures that credit assignment depends solely on the relative quality of a trajectory among its structural peers, rather than global statistics.

Despite these sharp conditional differences above, the following theorem establishes that both estimators are mathematically constrained to have identical global moments:

**Theorem 3.5** (Global Moments of SAN and GN). *Under the assumptions of Theorem 3.4, let $\varepsilon = 0$. The large-sample (population) advantages satisfy:*

*(a) Global Mean: $\mathbb{E}[A_{\text{SAN}} \mid x] = \mathbb{E}[A_{\text{GN}} \mid x] = 0$.*

*(b) Global Variance:*

$$\text{Var}(A_{\text{SAN}} \mid x) = \text{Var}(A_{\text{GN}} \mid x) = 1.$$

*(Proof in Appendix A.4).*

**Population-Level Gradient Consistency.** While SAN and GN share identical global moments, their *conditional* learning signals differ fundamentally, which governs the actual credit assignment. A potential concern is whether the stratification $S$ introduces spurious gradient bias at the population level. The following theorem resolves this by proving that the expected SAN gradient admits a clean decomposition: it exactly targets the weighted sum of the *true within-stratum policy gradients* $\nabla_\theta \mu_k$, scaled by their local stability. This confirms that SAN effectively eliminates the structural bias.

**Theorem 3.6** (Population SAN Expectation). *For a given prompt $x$, let $S = s(\tau, x)$ be a discrete stratum assignment determined by a fixed mapping $s$. Consider the population version of the SAN advantage $A_{\mathrm{SAN}}(\tau)$ defined in (2), where empirical moments are replaced by their corresponding expectations $\mu_k(\theta) = \mathbb{E}_\theta[R \mid S = k]$ and $\sigma_k^2(\theta) = \mathrm{Var}_\theta(R \mid S = k)$. Under some mild conditions,*

$$\mathbb{E}_{\tau \sim \pi_\theta}\left[A_{\mathrm{SAN}}(\tau)\,\nabla_\theta \log \pi_\theta(\tau)\right] = \sum_k \frac{p_k(\theta)}{\sigma_k(\theta) + \varepsilon}\,\nabla_\theta \mu_k(\theta), \quad (5)$$

*where $p_k(\theta) := \mathrm{Pr}_\theta(S = k)$. That is, the population SAN estimator targets a weighted sum of within-stratum gradients, scaled by their local stability.*

*(Detailed Proof in Appendix A.5).* The proof depends on Lemma A.2 in Appendix A.5. Given stratum $k$, the term $\nabla_\theta \mu_k(\theta)$ is the gradient of the expected reward conditioned on stratum $k$, the lemma gives the *advantage–score identity* conditional on $S = k$ as follows:

$$\nabla_\theta \mu_k(\theta) = \nabla_\theta \sum_{\tau \in B_k} R(\tau)\,\pi_\theta(\tau \mid S = k)$$

$$= \sum_{\tau \in B_k} R(\tau)\,\pi_\theta(\tau \mid S = k)\,\nabla_\theta \log \pi_\theta(\tau \mid S = k)$$

$$= \mathbb{E}_\theta\left[R(\tau)\,\nabla_\theta \log \pi_\theta(\tau \mid S = k)\,\big|\,S = k\right].$$

And, by $\mathbb{E}_\theta[\nabla_\theta \log \pi_\theta(\tau \mid S = k) \mid S = k] = 0$,

$$\nabla_\theta \mu_k(\theta)$$
$$= \mathbb{E}_{\tau \sim \pi_\theta}\left[(R(\tau) - \mu_k(\theta))\,\nabla_\theta \log \pi_\theta(\tau \mid S = k)\right].$$

This explicitly identifies $\nabla_\theta \mu_k(\theta)$ as the *true within-stratum policy gradient*. Consequently, the SAN estimator targets a weighted sum of the true within-stratum policy gradients, explicitly aligning the optimization direction with the reward landscape of each stratum.

Overall, SAN standardizes rewards within each stratum (Equation (2)), producing peer-consistent learning signals (Theorem 3.4). It admits a clear population interpretation as a weighted sum of true per-stratum gradients (Theorem 3.6) and avoids the conditional bias induced by global normalization (Theorem 3.4). Thus, SAN is a principled and stable estimator for learning under heterogeneous trajectories.

# 4. Blended Advantage for Finite-Sample Stability

The analysis in Theorems 3.4 and 3.6 characterizes the population and large-sample behavior of SAN, under which per-stratum normalization yields a pure and scale-stable learning signal. These guarantees, however, do not directly extend to finite-sample training, where some strata may be small or imbalanced, making advantage estimate noisy. To mitigate this effect, we introduce a linear blending of SAN with the global normalized (GN) estimator. This construction preserves SAN's local, peer-consistent structure while leveraging GN's global signal to stabilize learning under limited samples.

**Definition 4.1** (Blended Advantage). For $\tau \in B_k(x)$, define

$$A_{\mathrm{blend}}(\tau) = \alpha\,A_{\mathrm{SAN}}(\tau) + (1 - \alpha)\,A_{\mathrm{GN}}(\tau), \quad (6)$$

where $\alpha \in [0, 1]$.

The endpoints recover known estimators: $\alpha = 1$ yields SAN, and $\alpha = 0$ yields GN. Incorporating the blended advantage into SAN gives our practical method, *Stratified GRPO* (Algorithm 1)

# 5. Experiments

We evaluate our Stratified GRPO on factual QA tasks following Search-R1 (Jin et al., 2025) and on challenging deep-research agent tasks following VerlTool (Jiang et al., 2025). Furthermore, we provide extensive ablation studies and empirical analysis of our proposed Stratified GRPO.

## 5.1. Experiment Setup

**Models and Training.** For factual QA tasks, we conduct experiments on the Qwen-2.5-3B Base and Instruct models (Yang et al., 2024). For retrieval, we use the 2018 Wikipedia dump (Karpukhin et al., 2020) as the knowledge source and E5 (Wang et al., 2022) as the retriever, fetching the top-3 passages per query. Following the setup in Jin et al. (2025), we construct our training set by merging the training splits of Natural Questions (NQ) (Kwiatkowski et al., 2019) and HotpotQA (Yang et al., 2018). We use Exact Match (EM) as the training reward. For deep-research agent tasks, we follow the same settings in VerlTool (Jiang et al., 2025). Specifically, we use Qwen3-8B (Yang et al., 2025) as the base model and equip it with a Google search engine and a sandboxed Python executor as tools. We train the model using 1000 training samples from SimpleDeepSearcher (Sun et al., 2025) and Web-Sailor (Li et al., 2025a). Additional experiment details are in Appendix C.

**Stratification Criterion.** In all experiments, we stratify trajectories within each prompt based on the number of tool

*Table 1.* Experiment results on seven factual QA benchmarks. **Bold** denotes best results.

| Methods | Single-Hop QA | | | Multi-Hop QA | | | | Avg. |
|---|---|---|---|---|---|---|---|---|
| | NQ | TriviaQA | PopQA | HotpotQA | 2Wiki | Musique | Bamboogle | |
| *Non-RL Baselines* | | | | | | | | |
| Direct Generation | 10.6 | 28.8 | 10.8 | 14.9 | 24.4 | 2.0 | 2.4 | 13.4 |
| SFT | 24.9 | 29.2 | 10.4 | 18.6 | 24.8 | 4.4 | 11.2 | 17.6 |
| RAG | 34.8 | 54.4 | 38.7 | 25.5 | 22.6 | 4.7 | 8.0 | 27.0 |
| Search-o1 | 23.8 | 47.2 | 26.2 | 22.1 | 21.8 | 5.4 | 32.0 | 25.5 |
| IRCoT | 11.1 | 31.2 | 20.0 | 16.4 | 17.1 | 6.7 | 24.0 | 18.1 |
| *Qwen2.5-3B-Base* | | | | | | | | |
| Search-R1 | 40.6 | 58.7 | 43.5 | 28.4 | 27.3 | 4.9 | 8.8 | 30.3 |
| R1 | 22.6 | 45.5 | 17.3 | 20.1 | 26.8 | 5.5 | 22.4 | 22.9 |
| ReSearch | 42.7 | 59.7 | 43.0 | 30.5 | 27.2 | 7.4 | 12.8 | 31.9 |
| GRPO | 45.2 | 61.2 | **43.8** | 32.6 | 29.7 | 7.8 | 12.9 | 33.3 |
| **Stratified GRPO** | **45.9** | **61.4** | 43.0 | **40.8** | **39.9** | **17.7** | **42.7** | **41.6** |
| *Qwen2.5-3B-Instruct* | | | | | | | | |
| Search-R1 | 34.1 | 54.5 | 37.8 | 32.4 | 31.9 | 10.3 | 26.4 | 32.5 |
| R1 | 21.0 | 44.9 | 17.1 | 20.8 | 27.5 | 6.0 | 19.2 | 22.4 |
| ReSearch | 36.5 | 57.1 | 39.5 | 35.1 | 27.2 | 9.5 | 26.6 | 33.1 |
| GRPO | 33.4 | 52.9 | 36.7 | 26.5 | 27.4 | 6.4 | 21.0 | 29.2 |
| **Stratified GRPO** | **44.5** | **60.9** | **44.3** | **41.0** | **37.3** | **16.9** | **38.7** | **40.5** |

calls. This choice is motivated by our analysis of heterogeneity drivers in Section 3.2, which identifies the number of tool calls as the primary factor distinguishing trajectory structures and reward distributions. This stratification criterion ensures that the advantage is computed among structurally homogeneous peers.

**Evaluation Benchmarks.** For factual QA tasks, we evaluate performance on seven diverse question-answering datasets. These include three single-hop QA benchmarks: NQ (Kwiatkowski et al., 2019), TriviaQA (Joshi et al., 2017), and PopQA (Mallen et al., 2023); and four multi-hop QA benchmarks: HotpotQA (Yang et al., 2018), 2WikiMultiHopQA (Ho et al., 2020), MuSiQue (Trivedi et al., 2022), and Bamboogle (Press et al., 2023). Consistent with standard practice (Yu et al., 2024; Jin et al., 2025), EM is used as the evaluation metric. For deep-research agent tasks, we evaluate on the General AI Assistant (GAIA) benchmark (Mialon et al., 2024).

**Baselines.** For factual QA tasks, we compare Stratified GRPO against a comprehensive set of non-RL and RL methods. Non-RL methods include Direct Generation, Supervised Fine-Tuning (SFT), RAG (Lewis et al., 2020), Search-o1 (Li et al., 2025b), and IRCoT (Trivedi et al., 2023). RL Methods includes Search-R1 (Jin et al., 2025), RL without search (R1) (DeepSeek-AI et al., 2025), ReSearch (Chen et al., 2025), and GRPO (Shao et al., 2024). Most baseline results are cited from Jin et al. (2025), since their experiment setting is consistent with ours. For deep-research agent tasks,

we compare Stratified GRPO against large models without tools, agents with tool integrated reasoning, and GRPO. For large models without tools, we include Qwen3-32B-thinking (Yang et al., 2025), DeepSeek-R1-32B, DeepSeek-R1-671B (DeepSeek-AI et al., 2025), QwQ-32B (Team, 2025), and GPT-4o (Achiam et al., 2023) as baselines. For agents with tool integrated reasoning, we include RAG, Search-o1, WebThinker (Li et al., 2025c), and ReAct (Yao et al., 2023).

### 5.2. Main Results

**Results on Factual QA Tasks.** The factual QA experiment results, summarized in Table 1, demonstrate that Stratified GRPO consistently outperforms all baseline methods across seven QA benchmarks. On average, our method improves upon GRPO by up to 11.3 points and surpasses the best-performing baseline by up to 8.3 points. The advantage is particularly pronounced on multi-hop benchmarks, where Stratified GRPO achieves an average performance gain of up to 14.5 points over the strongest baseline. We attribute this success to our method's ability to eliminate systematic cross-stratum bias, enabling more effective learning from trajectories with varying search counts.

**Results on Deep-Research Agent Tasks.** To evaluate the effectiveness of Stratified GRPO on real-world multi-step tool-use scenarios, we assess its performance on the General AI Assistant (GAIA) benchmark (Mialon et al., 2024), a rigorous suite designed to test comprehensive capabilities in reasoning, web browsing, code execution, and tool pro-

*Table 2.* Performance on the GAIA benchmark. Stratified GRPO significantly outperforms baselines, particularly on complex tasks (Level 2 & 3), demonstrating superior performance in challenging real-world tool-use scenarios. **Bold** denotes best results.

| Method | Level 1 | Level 2 | Level 3 | Avg. |
|---|---|---|---|---|
| *Reasoning without Tool* | | | | |
| Qwen3-32B-thinking | 26.2 | 12.1 | 0.0 | 14.9 |
| DeepSeek-R1-32B | 21.5 | 13.6 | 0.0 | 14.2 |
| QwQ-32B | 30.9 | 6.5 | 5.2 | 18.9 |
| GPT-4o | 23.1 | 15.4 | 8.3 | 17.5 |
| DeepSeek-R1-671B | 40.5 | 21.2 | 5.2 | 25.2 |
| *Tool Integrated Reasoning (Qwen3-8B)* | | | | |
| Vanilla RAG | 28.2 | 15.4 | 16.7 | 20.4 |
| Search-o1 | 35.9 | 15.4 | 0.0 | 21.4 |
| WebThinker | 43.6 | 11.5 | 0.0 | 22.3 |
| ReAct | 35.9 | 17.3 | 8.3 | 23.3 |
| *RL-based Method* | | | | |
| Qwen3-8B | 28.1 | 15.4 | 16.7 | 20.4 |
| + GRPO | 48.7 | 32.7 | 16.7 | 36.9 |
| **+ Stratified GRPO** | **61.5** | **44.2** | **33.3** | **49.5** |

*Table 3.* Ablation study analyzing the components of Stratified GRPO, comparing the baseline GRPO, GRPO w/ SAN, and our full Stratified GRPO. TQA = TriviaQA, HP = HotpotQA, Mus = Musique, Bam = Bamboogle.

| Method | NQ | TQA | PopQA | HP | 2Wiki | Mus | Bam | Avg. |
|---|---|---|---|---|---|---|---|---|
| *Qwen2.5-3B-Base* | | | | | | | | |
| GRPO | 45.2 | 61.2 | **43.8** | 32.6 | 29.7 | 7.8 | 12.9 | 33.3 |
| w/ SAN | 43.7 | 59.3 | 41.1 | 36.6 | 38.4 | 12.6 | 25.0 | 36.7 |
| **Stratified GRPO** | **45.9** | **61.4** | 43.0 | **40.8** | **39.9** | **17.7** | **42.7** | **41.6** |
| *Qwen2.5-3B-Instruct* | | | | | | | | |
| GRPO | 33.4 | 52.9 | 36.7 | 26.5 | 27.4 | 6.4 | 21.0 | 29.2 |
| w/ SAN | 42.5 | 60.1 | 44.2 | 39.4 | **41.0** | 16.0 | 36.3 | 39.9 |
| **Stratified GRPO** | **44.5** | **60.9** | **44.3** | **41.0** | 37.3 | **16.9** | **38.7** | **40.5** |

*Table 4.* Validation of stratification criterion by comparing our stratification based on tool call numbers with random stratification.

| Method | NQ | TQA | PopQA | HP | 2Wiki | Mus | Bam | Avg. |
|---|---|---|---|---|---|---|---|---|
| GRPO | 45.2 | 61.2 | 43.8 | 32.6 | 29.7 | 7.8 | 12.9 | 33.3 |
| Random (2 strata) | 43.9 | 60.7 | 43.9 | 31.6 | 29.4 | 7.8 | 14.5 | 33.1 |
| Random (3 strata) | 44.1 | 61.4 | 43.6 | 32.5 | 28.0 | 7.7 | 15.3 | 33.2 |
| **Stratified GRPO** | 45.9 | 61.4 | 43.0 | 40.8 | 39.9 | 17.7 | 42.7 | 41.6 |

ficiency. Concretely, we train a Qwen3-8B agent equipped with Google search engine and python interpreter as tools and compare against a comprehensive set of strong large models and agent baselines.

Table 2 highlights the substantial gains achieved by Stratified GRPO, which outperforms the standard GRPO baseline by an average of **12.6 points**. Notably, this performance gap widens on the most challenging tasks (Level 3), where Stratified GRPO yields a **2×** **relative improvement** over GRPO (33.3 vs. 16.7). Overall, these results suggest that the benefits of mitigating cross-stratum comparisons are not limited to factual QA and can extend to a more complex multi-tool search-agent setting involving open-web search, code execution, and long-horizon reasoning.

### 5.3. Analysis

This section empirically validates the core mechanisms of our proposed Stratified GRPO. First, we conduct ablation studies to isolate the contribution of each component of Stratified GRPO. Next, we examine whether structure-aware stratification is necessary for the observed performance gains. We then assess the comparison and compatibility of Stratified GRPO with stronger RL training recipes. We further analyze the training dynamics of Stratified GRPO relative to GRPO, and how it enables the learning of effective multi-step search policies. Finally, we study the robustness of the blending coefficient. These results provide a comprehensive empirical confirmation of the theoretical analysis in Section 3.3. Further analysis of our Stratified GRPO is provided in Appendix D.

**Ablation Study.** We perform an ablation study to analyze the contribution of each component of our proposed Stratified GRPO. As shown in Table 3, each component provides a clear benefit. SAN alone yields significant gains over the baseline GRPO. The subsequent addition of advantage blending further enhances performance, establishing the effectiveness of the full Stratified GRPO algorithm, which consistently outperforms the other variants, especially on complex multi-hop QA tasks.

**Validation of the Stratification Strategy.** In Section 3.2, we identified the number of tool calls as the primary driver of heterogeneity in agents' tool-use trajectories, motivating its use as our stratification criterion. To validate this design choice, we conduct a control experiment where sampled rollouts are randomly assigned to two or three strata ($K = 2$ or 3), while keeping all other hyperparameters the same as our method. As shown in Table 4, random stratification yields performance nearly identical to the baseline GRPO (33.1–33.2 vs. 33.3 average). In contrast, our proposed Stratified GRPO improves the average by +8.3 points, with particularly significant gains on multi-hop datasets.

This result confirms that the performance benefits stem from structure-aware stratification rather than simple trajectory partitioning. Furthermore, this empirical finding aligns with our theoretical framework: when trajectories are assigned to strata randomly, stratum means and standard deviations converge to the global mean and global standard deviation respectively (up to sampling noise). Subsequently, by Proposition 3.3, random stratification fails to mitigate the cross-stratum bias, and the SAN advantage $A_{\text{SAN}}$ with random stratification degenerates to the GN advantage $A_{\text{GN}}$.

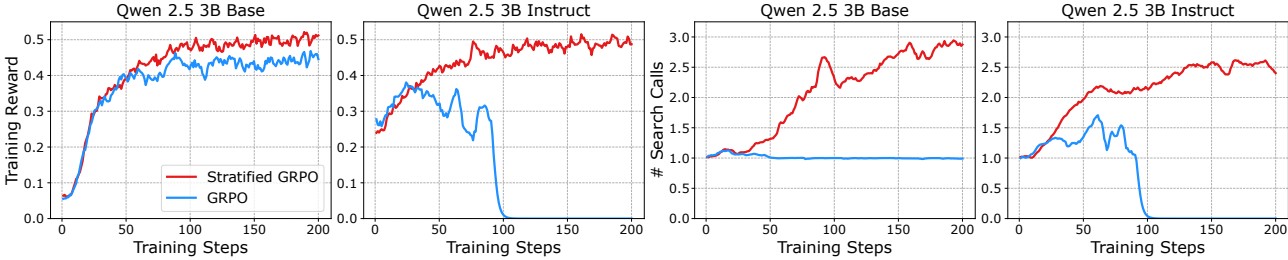

*Figure 1.* Training dynamics of Stratified GRPO and GRPO. The left plots show training rewards, and the right plots show the number of search calls per question over training steps.

*Table 5.* Comparison and compatibility with DAPO (Yu et al., 2025) on Qwen-2.5-3B-Base under the factual-QA setup.

| Method | NQ | TQA | PopQA | HP | 2Wiki | Mus | Bam | Avg. |
|---|---|---|---|---|---|---|---|---|
| GRPO | 45.2 | 61.2 | 43.8 | 32.6 | 29.7 | 7.8 | 12.9 | 33.3 |
| **Stratified GRPO** | 45.9 | 61.4 | 43.0 | 40.8 | 39.9 | 17.7 | **42.7** | 41.6 |
| DAPO | 46.9 | 62.3 | 45.9 | 34.0 | 31.2 | 8.5 | 19.4 | 35.5 |
| **DAPO w/ Ours** | **48.1** | **63.6** | **47.2** | **44.2** | **42.7** | **19.4** | 41.1 | **43.8** |

*Table 6.* Sensitivity analysis of the blending coefficient $\alpha$ on Qwen-2.5-3B-Base. Our method shows robust improvements over a wide range of $\alpha$, with the best performance achieved at $\alpha = 0.6$.

| Value of $\alpha$ | NQ | TQA | PopQA | HP | 2Wiki | Mus | Bam | Avg. |
|---|---|---|---|---|---|---|---|---|
| 0.0 (GRPO) | 45.2 | 61.2 | 43.8 | 32.6 | 29.7 | 7.8 | 12.9 | 33.3 |
| 0.2 | 45.1 | 61.8 | 43.1 | 32.2 | 29.5 | 7.6 | 14.5 | 33.4 |
| 0.4 | 44.6 | 59.8 | 43.2 | 39.6 | 36.5 | 15.6 | 39.5 | 39.8 |
| **0.6 (Default)** | **45.9** | 61.4 | 43.0 | **40.8** | 39.9 | **17.7** | **42.7** | **41.6** |
| 0.8 | 45.1 | 60.1 | **43.9** | 40.3 | **41.9** | 17.5 | 37.9 | 41.0 |
| 1.0 (SAN-only) | 43.7 | 59.3 | 41.1 | 36.6 | 38.4 | 12.6 | 25.0 | 36.7 |

**Comparison and Compatibility with Stronger RL Baselines.** To assess whether the gains of Stratified GRPO persist beyond the vanilla GRPO baseline, we compare it with DAPO (Yu et al., 2025), a stronger RL training recipe, under the same factual-QA setting on Qwen-2.5-3B-Base. We further evaluate whether our method can be combined with DAPO to test its compatibility with stronger RL baselines. As shown in Table 5, DAPO improves GRPO from 33.3 to 35.5 on average, while Stratified GRPO achieves a stronger average score of 41.6. Adding our method to DAPO further improves the average score to 43.8. These results suggest that Stratified GRPO addresses a complementary normalization issue and can be combined with stronger RL training recipes such as DAPO.

**Improved Training Reward and Stability.** Figure 1 (left) illustrates the training reward curves. For the base model, Stratified GRPO consistently achieves higher rewards than the standard GRPO baseline. More notably, when applied to the instruct model, standard GRPO suffers from training collapse, a known instability issue documented by prior works (Jin et al., 2025). In contrast, Stratified GRPO maintains a stable and monotonically increasing reward signal, demonstrating its superior stability and learning efficiency.

**Learning an Effective Search Policy.** A crucial ability for search agents is learning to identify knowledge gaps and issue search queries accordingly. We analyze this by tracking the average number of search calls per question during training (Figure 1, right). Stratified GRPO successfully learns a policy that converges to approximately 2.5 search calls, indicating it has learned to perform iterative searches. Conversely, the baseline GRPO stagnates at around one search call for the base model and causes training collapse for the

instruct model. This is because GRPO's cross-stratum bias prevents it from exploring more complex and potentially better multi-step search policies. This result demonstrates that Stratified GRPO learns a more effective search policy, which directly translates to its superior performance on multi-hop benchmarks that require sequential information retrieval, as shown in Table 1.

**Robustness of Blending Coefficient $\alpha$.** In Section 4, we introduced the blending coefficient $\alpha$ to leverage GN's global signal to stabilize learning under limited samples. To verify the robustness of this hyperparameter, we conduct a sweep over $\alpha \in \{0.0, 0.2, 0.4, 0.6, 0.8, 1.0\}$ on Qwen-2.5-3B-Base. The results in Table 6 yield two key observations. (1) **Robust Improvement:** Stratified GRPO consistently outperforms the baseline GRPO ($\alpha = 0$) across all non-zero values. A high-performance plateau exists between $\alpha = 0.4$ and $\alpha = 0.8$, indicating that the method is not overly sensitive to the exact choice of $\alpha$. (2) **Benefit of Blending:** While the pure SAN variant ($\alpha = 1.0$) already achieves substantial gains over GRPO (36.7 vs. 33.3), the blended approach yields the best overall performance (41.6 at $\alpha = 0.6$). This empirically validates our theoretical motivation: blending preserves SAN's stratification while leveraging the global signal to stabilize finite-sample estimation.

## 6. Conclusion

In this work, we identify and formalize cross-stratum bias, a key obstacle for training LLM search agents with RL. It arises from improperly comparing structurally heterogeneous trajectories using a global baseline. It causes distorted

credit assignment and hampers exploration. To address this, we introduce Stratified GRPO, a principled algorithm that partitions trajectories into homogeneous strata and computes advantages locally. Our analysis proves this method eliminates cross-stratum bias and achieves conditional unbiasedness and unit variance within each stratum, while retaining these properties globally. Extensive experiments on diverse QA and deep-research benchmarks demonstrate that Stratified GRPO substantially outperforms GRPO by up to 12.6 points, achieving higher training rewards, greater training stability, and more effective search strategies. These results support structure-aware advantage normalization as an effective correction for GRPO-style training of search agents with structurally heterogeneous rollouts.

## Acknowledgements

This work was supported in part by the Research Grants Council under the Areas of Excellence scheme grant AoE/E-601/22-R.

## Impact Statement

This work aims to improve reinforcement learning for search and tool-use agents. While stronger search agents can help with complex information-seeking tasks, but they may also retrieve unreliable, or sensitive information and propagate such errors through multi-step reasoning. Practical deployment should include source-quality checks, usage monitoring, and task-specific safety constraints.

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

---

**Algorithm 1** Stratified GRPO

---

**Require:** Policy $\pi_\theta$, batch $B = \{\tau_1, \ldots, \tau_K\}$ with rewards $\{R_i\}$, blending $\alpha \in [0, 1]$, stabilizer $\varepsilon > 0$.

1: Compute global stats: $\bar{R}_{\text{global}} \leftarrow \frac{1}{K} \sum_i R_i \quad \widehat{\sigma}_{\text{global}} \leftarrow \sqrt{\frac{1}{K} \sum_i (R_i - \bar{R}_{\text{global}})^2}$.

2: For all $i$, set $A_{\text{GN}}(\tau_i) \leftarrow (R_i - \bar{R}_{\text{global}})/(\widehat{\sigma}_{\text{global}} + \varepsilon)$.

3: Partition indices into per-prompt, per-stratum groups $I_k(x)$ (e.g., by search count).

4: **for** each prompt $x$ **do**

5:     **for** each stratum $k$ with index set $I_k(x)$ **do**

6:         $n_k \leftarrow |I_k(x)|; \quad \widehat{\mu}_k \leftarrow \frac{1}{n_k} \sum_{i \in I_k(x)} R_i; \quad \widehat{\sigma}_k \leftarrow \sqrt{\frac{1}{n_k} \sum_{i \in I_k(x)} (R_i - \widehat{\mu}_k)^2}$.

7:         **for** $i \in I_k(x)$ **do**

8:             $A_{\text{SAN}}(\tau_i) \leftarrow (R_i - \widehat{\mu}_k)/(\widehat{\sigma}_k + \varepsilon)$.

9:             $A_{\text{blend}}(\tau_i) \leftarrow \alpha A_{\text{SAN}}(\tau_i) + (1 - \alpha) A_{\text{GN}}(\tau_i)$.

10:         **end for**

11:     **end for**

12: **end for**

13: Return gradient estimate $\widehat{g}_{\text{blend}} \leftarrow \frac{1}{K} \sum_{i=1}^{K} A_{\text{blend}}(\tau_i) \nabla_\theta \log \pi_\theta(\tau_i \mid x)$.

---

# A. Proofs of Theoretical Results

## A.1. Invariance to Positive Affine Reward Transforms of $A_{\text{SAN}}$

**Proposition A.1** (Invariance to Positive Affine Reward Transforms). *Suppose $\varepsilon = 0$. The SAN advantage $A_{SAN}(\tau)$ is invariant under any positive affine transformation of the rewards, $R'(\tau) = aR(\tau) + b$ with $a > 0$. That is, $A'_{SAN}(\tau) = A_{SAN}(\tau)$.*

*Proof.* Let a stratum for a given prompt $x$ be denoted by $B_k(x)$, containing $n_k(x)$ trajectories with rewards $\{R_1, \ldots, R_{n_k(x)}\}$. The empirical mean and standard deviation are

$$\widehat{\mu}_k(x) = \frac{1}{n_k(x)} \sum_{i=1}^{n_k(x)} R_i$$

and

$$\widehat{\sigma}_k(x) = \sqrt{\frac{1}{n_k(x)} \sum_{i=1}^{n_k(x)} (R_i - \widehat{\mu}_k(x))^2}.$$

Without loss of generality, suppose that $n_k \geq 2$, and $\widehat{\sigma}_k(x) > 0$.

Consider the affine transformation $R'_i = aR_i + b$ for $a > 0$. We first compute the new empirical mean $\widehat{\mu}'_k(x)$ and standard deviation $\widehat{\sigma}'_k(x)$ for the transformed rewards.

The new mean is:

$$\widehat{\mu}'_k(x) = \frac{1}{n_k(x)} \sum_{i=1}^{n_k(x)} R'_i = \frac{1}{n_k(x)} \sum_{i=1}^{n_k(x)} (aR_i + b) = a \left( \frac{1}{n_k(x)} \sum_{i=1}^{n_k(x)} R_i \right) + b = a\widehat{\mu}_k(x) + b.$$

The new variance is:

$$(\widehat{\sigma}'_k(x))^2 = \frac{1}{n_k(x)} \sum_{i=1}^{n_k(x)} (R'_i - \widehat{\mu}'_k(x))^2$$

$$= \frac{1}{n_k(x)} \sum_{i=1}^{n_k(x)} ((aR_i + b) - (a\widehat{\mu}_k(x) + b))^2$$

$$= \frac{1}{n_k(x)} \sum_{i=1}^{n_k(x)} \big(a(R_i - \widehat{\mu}_k(x))\big)^2$$

$$= a^2 \left( \frac{1}{n_k(x)} \sum_{i=1}^{n_k(x)} (R_i - \widehat{\mu}_k(x))^2 \right) = a^2 (\widehat{\sigma}_k(x))^2.$$

Since $a > 0$, the new standard deviation is $\widehat{\sigma}'_k(x) = \sqrt{a^2(\widehat{\sigma}_k(x))^2} = a\widehat{\sigma}_k(x)$.

Finally, we compute the new advantage $A'_{\text{SAN}}$ for an arbitrary trajectory $\tau_i \in B_k(x)$:

$$\begin{aligned}
A'_{\text{SAN}}(\tau_i) &= \frac{R'_i - \widehat{\mu}'_k(x)}{\widehat{\sigma}'_k(x)} \\
&= \frac{(aR_i + b) - (a\widehat{\mu}_k(x) + b)}{a\widehat{\sigma}_k(x)} \\
&= \frac{a(R_i - \widehat{\mu}_k(x))}{a\widehat{\sigma}_k(x)} \\
&= \frac{R_i - \widehat{\mu}_k(x)}{\widehat{\sigma}_k(x)} = A_{\text{SAN}}(\tau_i).
\end{aligned}$$

This shows that the advantage computed from the transformed rewards is identical to the original, completing the proof. $\square$

## A.2. Proof of Proposition 3.3

*Proof.* The proof follows from direct algebraic manipulation. We start from the right-hand side (RHS) of Equation (3) and substitute the definitions of $\alpha_k(x)$, $A_{\text{SAN}}(\tau_i)$, and $\Delta_k(x)$:

$$\begin{aligned}
\text{RHS} &= \alpha_k(x) \cdot A_{\text{SAN}}(\tau_i) + \Delta_k(x) \\
&= \left( \frac{\widehat{\sigma}_k(x) + \varepsilon}{\widehat{\sigma}_{\text{global}}(x) + \varepsilon} \right) \cdot \left( \frac{R(\tau_i) - \widehat{\mu}_k(x)}{\widehat{\sigma}_k(x) + \varepsilon} \right) + \left( \frac{\widehat{\mu}_k(x) - \bar{R}_{\text{global}}(x)}{\widehat{\sigma}_{\text{global}}(x) + \varepsilon} \right) &&\text{(by definition)} \\
&= \frac{R(\tau_i) - \widehat{\mu}_k(x)}{\widehat{\sigma}_{\text{global}}(x) + \varepsilon} + \frac{\widehat{\mu}_k(x) - \bar{R}_{\text{global}}(x)}{\widehat{\sigma}_{\text{global}}(x) + \varepsilon} &&\text{(cancel } (\widehat{\sigma}_k(x) + \varepsilon)) \\
&= \frac{(R(\tau_i) - \widehat{\mu}_k(x)) + (\widehat{\mu}_k(x) - \bar{R}_{\text{global}}(x))}{\widehat{\sigma}_{\text{global}}(x) + \varepsilon} &&\text{(combine terms)} \\
&= \frac{R(\tau_i) - \bar{R}_{\text{global}}(x)}{\widehat{\sigma}_{\text{global}}(x) + \varepsilon} &&\text{(cancel } \widehat{\mu}_k(x)) \\
&= A_{\text{GN}}(\tau_i). &&\text{(definition of } A_{\text{GN}}) \qquad \square
\end{aligned}$$

## A.3. Proof of Theorem 3.4

*Proof.* The proof proceeds by direct calculation of the conditional expectation and variance for each estimator in the large-sample limit. All quantities are conditioned on a fixed prompt $x$.

First, we prove the results for $A_{\text{SAN}}$. The SAN advantage for a trajectory $\tau$ in stratum $k$ is defined as $A_{\text{SAN}}(\tau) = \frac{R(\tau) - \mu_k}{\sigma_k}$ (ignoring $\varepsilon$ for clarity in the limit). By the linearity of expectation,

$$\begin{aligned}
\mathbb{E}[A_{\text{SAN}}(\tau) \mid k, x] &= \mathbb{E}\left[ \frac{R(\tau) - \mu_k}{\sigma_k} \,\Big|\, k, x \right] \\
&= \frac{1}{\sigma_k} \big( \mathbb{E}[R(\tau) \mid k, x] - \mu_k \big)
\end{aligned}$$

$$= \frac{1}{\sigma_k}(\mu_k - \mu_k) = 0.$$

This shows that SAN is an unbiased signal carrier within each stratum. Next, by the properties of variance, $\mathrm{Var}(cX + d) = c^2 \mathrm{Var}(X)$,

$$\mathrm{Var}(A_{\mathrm{SAN}}(\tau) \mid k, x) = \mathrm{Var}\left(\frac{R(\tau) - \mu_k}{\sigma_k} \;\middle|\; k, x\right)$$
$$= \frac{1}{\sigma_k^2} \mathrm{Var}(R(\tau) - \mu_k \mid k, x)$$
$$= \frac{1}{\sigma_k^2} \mathrm{Var}(R(\tau) \mid k, x)$$
$$= \frac{\sigma_k^2}{\sigma_k^2} = 1.$$

This shows that SAN provides a consistently scaled (unit variance) carrier in every stratum.

Second, we prove the results for $A_{\mathrm{GN}}$. The GN advantage is defined as $A_{\mathrm{GN}}(\tau) = \frac{R(\tau) - \mu}{\sigma}$. So

$$\mathbb{E}[A_{\mathrm{GN}}(\tau) \mid k, x] = \mathbb{E}\left[\frac{R(\tau) - \mu}{\sigma} \;\middle|\; k, x\right]$$
$$= \frac{1}{\sigma}\left(\mathbb{E}[R(\tau) \mid k, x] - \mu\right)$$
$$= \frac{\mu_k - \mu}{\sigma}.$$

This is generally non-zero whenever stratum means differ ($\mu_k \neq \mu$), which confirms that GN is a biased carrier within the stratum. This non-zero expectation represents a spurious signal. Next,

$$\mathrm{Var}(A_{\mathrm{GN}}(\tau) \mid k, x) = \mathrm{Var}\left(\frac{R(\tau) - \mu}{\sigma} \;\middle|\; k, x\right)$$
$$= \frac{1}{\sigma^2} \mathrm{Var}(R(\tau) - \mu \mid k, x)$$
$$= \frac{1}{\sigma^2} \mathrm{Var}(R(\tau) \mid k, x) = \frac{\sigma_k^2}{\sigma^2}.$$

This shows that the variance of the GN carrier is not consistent across strata; it depends on the ratio of the stratum's variance to the global variance.

The analysis of the conditional statistics reveals the structural superiority of SAN. Within any given stratum, SAN provides a zero-mean, unit-variance signal carrier, which serves as a pure and consistent baseline for the policy gradient calculation. GN, in contrast, introduces a spurious signal (a non-zero mean) and has an inconsistent variance, making it a biased and less reliable signal carrier. This completes the proof. $\qquad\square$

### A.4. Proof of Theorem 3.5

*Proof.* Fix a prompt $x$ and let $S \in \{0, \ldots, I - 1\}$ denote the stratum index with mixing weights $p_k(x) = \Pr(S = k \mid x)$.

**For (a)**, applying the law of total expectation,

$$\mathbb{E}[A_{\mathrm{SAN}} \mid x] = \sum_k p_k(x)\, \mathbb{E}\left[\frac{R - \mu_k(x)}{\sigma_k(x)} \;\middle|\; S=k, x\right]$$
$$= \sum_k p_k(x)\, \frac{\mathbb{E}[R - \mu_k(x) \mid S=k, x]}{\sigma_k(x)}$$
$$= 0.$$

$$\mathbb{E}[A_{\text{GN}} \mid x] = \frac{\mathbb{E}[R \mid x] - \mu(x)}{\sigma(x)} = 0.$$

**For (b)**, by the law of total variance,

$$\text{Var}(A_{\text{SAN}} \mid x) = \mathbb{E}[\text{Var}(A_{\text{SAN}} \mid S, x)] + \text{Var}(\mathbb{E}[A_{\text{SAN}} \mid S, x]).$$

We now analyze each of the two terms in the right-hand side separately.

From Theorem 3.4, we know that the idealized SAN advantage is conditionally unbiased in every stratum:

$$\mathbb{E}[A_{\text{SAN}} \mid S = k, x] = 0, \quad \text{for all strata } k.$$

Since the conditional expectation is a constant (zero) regardless of the stratum $S$, its variance is zero:

$$\text{Var}(\mathbb{E}[A_{\text{SAN}} \mid S, x]) = \text{Var}(0) = 0.$$

Next, we evaluate the first term, which is the expected variance within strata. We start with the inner term, the variance conditional on a specific stratum $k$:

$$\text{Var}(A_{\text{SAN}} \mid S = k, x) = \text{Var}\left( \frac{R - \mu_k(x)}{\sigma_k(x)} \,\middle|\, k, x \right)$$

$$= \frac{1}{\sigma_k^2(x)} \text{Var}(R \mid k, x)$$

$$= \frac{\sigma_k^2(x)}{\sigma_k^2(x)} = 1.$$

Now, we take the expectation of this quantity over the random stratum $S$. This is equivalent to a weighted average over all strata, where the weights are the probabilities $p_k(x) := \Pr(S = k \mid x)$:

$$\mathbb{E}[\text{Var}(A_{\text{SAN}} \mid S, x)] = \sum_k p_k(x) \cdot \text{Var}(A_{\text{SAN}} \mid S = k, x)$$

$$= \sum_k p_k(x) \frac{\sigma_k^2(x)}{\sigma_k^2(x)} = \sum_k p_k(x) = 1.$$

Finally, we add the two components back together:

$$\text{Var}(A_{\text{SAN}} \mid x) = \mathbb{E}[\text{Var}(A_{\text{SAN}} \mid S, x)] + \text{Var}(\mathbb{E}[A_{\text{SAN}} \mid S, x]) = 1 + 0 = 1.$$

For GN, since $A_{\text{GN}} = (R - \mu(x))/\sigma(x)$,

$$\text{Var}(A_{\text{GN}} \mid x) = \frac{\text{Var}(R \mid x)}{\sigma^2(x)} = \frac{\sigma^2(x)}{\sigma^2(x)} = 1. \qquad \square$$

## A.5. Proof of Theorem 3.6

**Fixed-Support Differentiation Regime.** In our implementation, strata are defined by fixed, rule-based criteria (e.g., the number of tool calls), and each trajectory $\tau$ is assigned to a stratum $S = s(\tau, x)$ deterministically. When computing policy gradients, we do not backpropagate through this assignment; gradients are taken only with respect to the trajectory likelihood, treating the stratum index as fixed. Consequently, this convention is adopted throughout the following theoretical analysis.

For a fixed prompt $x$, the population statistics $\mu_k(x, \theta)$ and $\sigma_k(x, \theta)$ are defined over the support of all potential trajectories $\tau$ induced by the policy $\pi_\theta(\cdot \mid x)$ within stratum $k$. Specifically, let $B_k(x) = \{\tau : s(\tau, x) = k\}$ denote the set of trajectories

assigned to stratum $k$. The population mean is thus the conditional expectation (suppose $p_k(x, \theta) > 0$ without loss of generality):

$$\mu_k(x, \theta) = \mathbb{E}_{\tau \sim \pi_\theta(\cdot|x)}\big[R(\tau) \mid \tau \in B_k(x)\big] = \sum_{\tau \in B_k(x)} R(\tau) \frac{\pi_\theta(\tau|x)}{p_k(x, \theta)}.$$

To streamline the derivation, we consider a fixed prompt $x$ and suppress it in the notation where the context is clear, and use the notation $\mathbb{E}_\theta[\cdot]$ to denote $\mathbb{E}_{\tau \sim \pi_\theta(\cdot|x)}[\cdot]$ which is dependent on the variable $\theta$.

Proving Theorem 3.6 relies on the following lemma:

**Lemma A.2** (Conditional Advantage–Score Identity). *For any stratum $k$, the following identity holds:*

$$\mathbb{E}_\theta\big[(R(\tau) - \mu_k(\theta)) \nabla_\theta \log \pi_\theta(\tau \mid S = k) \mid S = k\big] = \nabla_\theta \mu_k(\theta),$$

*where $\mu_k(\theta) := \mathbb{E}_\theta[R(\tau) \mid S = k]$ is the mean reward in stratum $k$. This identity is derived under the Fixed-Support Differentiation Regime, where the gradient is taken with respect to the conditional policy likelihood only, treating the stratum boundaries as fixed. Moreover, we assume that $\pi_\theta(\tau \mid x)$ is differentiable in $\theta$, and that differentiation may be interchanged with summation/integration.*

*Proof.* By definition, the conditional reward mean can be written as (we assume $p_k(\theta) > 0$ without loss of generality):

$$\mu_k(\theta) = \mathbb{E}_\theta[R(\tau) \mid S = k] = \sum_{\tau \in B_k} R(\tau) \frac{\pi_\theta(\tau)}{p_k(\theta)},$$

where $B_k = \{\tau : S(\tau) = k\}$ and $p_k(\theta) = \Pr_\theta(S = k)$. Equivalently, this defines the induced conditional policy

$$\pi_\theta(\tau \mid S = k) := \frac{\pi_\theta(\tau)}{p_k(\theta)}, \qquad \tau \in B_k,$$

under which the expectation may be written as

$$\mu_k(\theta) = \sum_{\tau \in B_k} \pi_\theta(\tau \mid S = k)\, R(\tau).$$

Under the Fixed-Support Differentiation Regime, the stratum assignment is treated as fixed when differentiating with respect to $\theta$. Differentiating the expression above yields

$$\begin{aligned}
\nabla_\theta \mu_k(\theta) &= \nabla_\theta \sum_{\tau \in B_k} \pi_\theta(\tau \mid S = k)R(\tau) \\
&= \sum_{\tau \in B_k} R(\tau)\nabla_\theta \pi_\theta(\tau \mid S = k) \\
&= \sum_{\tau \in B_k} R(\tau)\pi_\theta(\tau \mid S = k)\nabla_\theta \log \pi_\theta(\tau \mid S = k) \\
&= \mathbb{E}_\theta[R(\tau) \nabla_\theta \log \pi_\theta(\tau \mid S = k) \mid S = k].
\end{aligned}$$

Next, we decompose the reward as $R(\tau) = (R(\tau) - \mu_k(\theta)) + \mu_k(\theta)$. Noting that $\mu_k(\theta)$ is constant with respect to the conditional expectation over $\tau$ for a given stratum $k$, we can factor it out of the second term:

$$\begin{aligned}
\nabla_\theta \mu_k(\theta) = \,&\mathbb{E}_\theta[(R(\tau) - \mu_k(\theta)) \nabla_\theta \log \pi_\theta(\tau \mid S = k) \mid S = k] \\
&+ \mu_k(\theta) \mathbb{E}_\theta[\nabla_\theta \log \pi_\theta(\tau \mid S = k) \mid S = k].
\end{aligned} \tag{7}$$

Under the Fixed-Support Differentiation Regime, strata are defined by fixed, rule-based criteria, and each trajectory $\tau$ is deterministically assigned to a stratum $S$. Although the induced stratum probabilities $p_k(\theta) = \Pr_\theta(S = k)$ generally depend

on $\theta$, gradients in the following analysis are taken without backpropagating through the stratum assignment. Consequently, the gradient operator acts only on the conditional trajectory likelihood, and the conditional score has zero mean:

$$
\begin{aligned}
\mathbb{E}_\theta[\nabla_\theta \log \pi_\theta(\tau \mid S = k) \mid S = k] &= \sum_{\tau \in B_k} \pi_\theta(\tau \mid S = k)\, \nabla_\theta \log \pi_\theta(\tau \mid S = k) \\
&= \sum_{\tau \in B_k} \nabla_\theta \pi_\theta(\tau \mid S = k) \qquad \text{(since } \pi \nabla \log \pi = \nabla \pi \text{)} \\
&= \nabla_\theta \sum_{\tau \in B_k} \pi_\theta(\tau \mid S = k) \\
&= \nabla_\theta \sum_{\tau \in B_k} \frac{\pi_\theta(\tau)}{p_k(\theta)} \\
&= \nabla_\theta(1) \\
&= 0.
\end{aligned}
$$

Substituting this into Equation (7) yields the stated identity. $\qquad\square$

Next, we give a proof of Theorem 3.6 under the assumptions of Lemma A.2:

*Proof.* We prove the identity by first applying the law of total expectation to decompose the total expectation over the strata $k$:

$$
\begin{aligned}
\mathbb{E}_\theta\big[A_{\mathrm{SAN}}(\tau)\, \nabla_\theta \log \pi_\theta(\tau)\big] &= \mathbb{E}_\theta\Big[A_{\mathrm{SAN}}(\tau)\, \nabla_\theta \log \pi_\theta(\tau) \sum_k \mathbf{1}_{S=k}\Big] \qquad (\text{by } \sum_k \mathbf{1}_{S=k} = 1) \\
&= \sum_k \mathbb{E}_\theta\Big[A_{\mathrm{SAN}}(\tau)\, \nabla_\theta \log \pi_\theta(\tau)\, \mathbf{1}_{S=k}\Big] \\
&= \sum_k \mathbb{E}_\theta\Big[\mathbf{1}_{S=k}\, \mathbb{E}_\theta\big[A_{\mathrm{SAN}}(\tau)\, \nabla_\theta \log \pi_\theta(\tau) \,\big|\, S = k\big]\Big] \\
&= \sum_k \underbrace{\mathbb{E}_\theta[\mathbf{1}_{S=k}]}_{=\, p_k(\theta)}\, \mathbb{E}_\theta\big[A_{\mathrm{SAN}}(\tau)\, \nabla_\theta \log \pi_\theta(\tau) \,\big|\, S = k\big] \\
&= \sum_k p_k(\theta)\, \mathbb{E}_\theta\big[A_{\mathrm{SAN}}(\tau)\, \nabla_\theta \log \pi_\theta(\tau) \,\big|\, S = k\big].
\end{aligned} \tag{8}
$$

Now, we analyze the conditional expectation for a single stratum $k$. The key is to decompose the score function using the chain rule of probability, $\pi_\theta(\tau) = \pi_\theta(\tau \mid S = k)\, p_k(\theta)$. Taking the log-gradient gives the identity:

$$
\nabla_\theta \log \pi_\theta(\tau) = \nabla_\theta \log \pi_\theta(\tau \mid S = k) + \nabla_\theta \log p_k(\theta).
$$

Substituting this into the conditional expectation allows us to split it into two terms by linearity:

$$
\begin{aligned}
\mathbb{E}_\theta\Big[&A_{\mathrm{SAN}}(\tau)\, \nabla_\theta \log \pi_\theta(\tau) \,\big|\, S = k\Big] \\
&= \underbrace{\mathbb{E}_\theta\Big[A_{\mathrm{SAN}}(\tau)\, \nabla_\theta \log \pi_\theta(\tau \mid S = k) \,\big|\, S = k\Big]}_{\text{Term 1: Conditional Score Part}} + \underbrace{\mathbb{E}_\theta\Big[A_{\mathrm{SAN}}(\tau)\, \nabla_\theta \log p_k(\theta) \,\big|\, S = k\Big]}_{\text{Term 2: Marginal Score Part}}.
\end{aligned} \tag{9}
$$

We analyze each term separately.

**Term 2 (Marginal Score Part).** The term $\nabla_\theta \log p_k(\theta)$ depends only on the stratum index $k$ and the parameter $\theta$, and is therefore constant with respect to the conditional expectation given $S = k$. We may thus factor it out:

$$
\mathbb{E}_\theta\Big[A_{\mathrm{SAN}}(\tau)\, \nabla_\theta \log p_k(\theta) \,\big|\, S = k\Big] = (\nabla_\theta \log p_k(\theta)) \cdot \mathbb{E}_\theta[A_{\mathrm{SAN}}(\tau) \mid S = k]
$$

$$= (\nabla_\theta \log p_k(\theta)) \cdot \frac{\mathbb{E}_\theta[R(\tau) - \mu_k(\theta) \mid S = k]}{\sigma_k(\theta) + \varepsilon}$$

$$= (\nabla_\theta \log p_k(\theta)) \cdot \frac{\mu_k(\theta) - \mu_k(\theta)}{\sigma_k(\theta) + \varepsilon} = 0.$$

Thus, the second term vanishes exactly. This is the crucial step where the structural part of the gradient is eliminated.

**Term 1 (Conditional score part).** For this term, we substitute the definition of $A_{\mathrm{SAN}}(\tau)$. Conditioned on $S = k$,

$$\mathbb{E}_\theta \Big[ A_{\mathrm{SAN}}(\tau) \, \nabla_\theta \log \pi_\theta(\tau \mid S = k) \,\Big|\, S = k \Big]$$

$$= \mathbb{E}_\theta \left[ \frac{R(\tau) - \mu_k(\theta)}{\sigma_k(\theta) + \varepsilon} \, \nabla_\theta \log \pi_\theta(\tau \mid S = k) \,\bigg|\, S = k \right]$$

$$= \frac{1}{\sigma_k(\theta) + \varepsilon} \, \mathbb{E}_\theta \Big[ (R(\tau) - \mu_k(\theta)) \, \nabla_\theta \log \pi_\theta(\tau \mid S = k) \,\big|\, S = k \Big].$$

Under the Fixed-Support Differentiation Regime, in the above equation, gradients are taken only with respect to the conditional policy likelihood $\pi_\theta(\tau \mid S = k)$. Applying the Conditional Advantage–Score Identity (Lemma A.2), the remaining expectation equals $\nabla_\theta \mu_k(\theta)$. Hence,

$$\mathbb{E}_\theta \Big[ A_{\mathrm{SAN}}(\tau) \, \nabla_\theta \log \pi_\theta(\tau \mid S = k) \,\big|\, S = k \Big] = \frac{1}{\sigma_k(\theta) + \varepsilon} \, \nabla_\theta \mu_k(\theta).$$

**Conclusion.** Substituting the results for Term 1 and Term 2 back into Equation (9), and by the original sum over strata in Equation (8), we obtain the final result:

$$\mathbb{E}_\theta \big[ A_{\mathrm{SAN}}(\tau) \, \nabla_\theta \log \pi_\theta(\tau) \big] = \sum_k p_k(\theta) \left( \frac{1}{\sigma_k(\theta) + \varepsilon} \, \nabla_\theta \mu_k(\theta) + 0 \right) = \sum_k \frac{p_k(\theta)}{\sigma_k(\theta) + \varepsilon} \, \nabla_\theta \mu_k(\theta).$$

This completes the proof. $\qquad\square$

## B. Comparison of Local vs. Global Moments of SAN and GN Advantages

*Table 7.* Local (conditional on stratum $S{=}k$) vs. global (marginal over $S$) moments of SAN and GN advantages. Here $\sigma_k^2 = \mathrm{Var}(R \mid S{=}k, x)$ and $\sigma^2 = \mathrm{Var}(R \mid x)$. For $\varepsilon = 0$, SAN achieves joint standardization (local & global), whereas GN only standardizes globally.

|  | **Local (given $S{=}k$)** | **Global (marginal over $S$)** |
|---|:---:|:---:|
| SAN mean | 0 | 0 |
| SAN variance | 1 | 1 |
| GN mean | $\frac{\mu_k - \mu}{\sigma}$ | 0 |
| GN variance | $\frac{\sigma_k^2}{\sigma^2}$ | 1 |

## C. Experiment Details: Training Settings

For factual QA tasks, we mainly follow the settings of Search-R1 (Jin et al., 2025). We train our models on 8 GPUs using a global batch size of 256 and a mini-batch size of 256. The maximum sequence length is set to 4096 tokens, with maximum response length and retrieved content length of 500 tokens in each interaction turn. For rollout sampling, we use a temperature of 1.0 and a top-$p$ value of 1.0. We use a learning rate of 1e-6 with a warm-up ratio of 0.1. Training is conducted for 200 steps. We use a KL divergence coefficient $\beta$ of 0.001 and a clipping ratio $\epsilon$ of 0.2. For both GRPO and Stratified GRPO, we sample 8 responses per prompt. Stratified GRPO uses $\alpha$ of 0.8 for Qwen 2.5 3B Instruct and 0.6 for Qwen 2.5 3B Base. The number of maximum interaction turns is set to 4, and we retrieve the top 3 passages for each search call. Our implementation is based on the Verl framework (Sheng et al., 2025). For deep-research agent tasks, we mainly follow the settings of VerlTool (Jiang et al., 2025) and base our implementation on the VerlTool framework. Specifically, we

use a global batch size of 128 and a mini-batch size of 16. The maximum sequence length is set to 8192 tokens, and the number of maximum interaction turns is set to 5 during training. We use a learning rate of 1e-6 and sample 16 responses per prompt for GRPO and Stratified GRPO. Stratified GRPO uses $\alpha$ of 0.5. During evaluation, the maximum sequence length is set to 32768 tokens, and the number of maximum interaction turns is set to 10.

# D. Further Analysis of Stratified GRPO

## D.1. Multi-Seed Stability Analysis

We evaluate the robustness and stability of Stratified GRPO by conducting the factual QA experiment on Qwen-2.5-3B Base using three random seeds. As shown in Table 8, Stratified GRPO yields consistent gains across all seeds, underscoring the method's reliability.

We visualize the training dynamics averaged across these independent runs in Figure 2. Figure 2 demonstrates that Stratified GRPO achieves consistently higher training rewards and effective multi-turn search policies, alongside lower and smoother gradient norms and KL divergence compared to GRPO. Furthermore, we observe that Stratified GRPO successfully expands the average rollout length, whereas GRPO collapses to short responses, aligning with the "Search Calls" plot and confirming that our method encourages long-horizon exploration. While entropy converges at a similar rate for both methods, Stratified GRPO converges to a higher-reward policy. Regarding advantage estimates, the global advantage shows unit variance for both methods, supporting Theorem 3.5. Crucially, the in-stratum advantage variance confirms that Stratified GRPO maintains unit variance within each stratum while GRPO does not, empirically validating Theorem 3.4. Furthermore, Stratified GRPO's in-stratum advantage variance is consistently lower than GRPO, further demonstrating our method's stability.

*Table 8.* Stability analysis by training Qwen2.5-3B-Base using 3 random seeds for GRPO and our Stratified GRPO.

| Method | NQ | TriviaQA | PopQA | HotpotQA | 2Wiki | Musique | Bamboogle | Avg. |
|---|---|---|---|---|---|---|---|---|
| GRPO | $44.5 \pm 1.0$ | $60.9 \pm 0.4$ | $43.4 \pm 0.4$ | $32.1 \pm 0.5$ | $29.2 \pm 0.4$ | $8.0 \pm 0.2$ | $14.0 \pm 0.9$ | $33.2 \pm 0.2$ |
| **Stratified GRPO** | $45.8 \pm 0.2$ | $61.6 \pm 0.6$ | $44.0 \pm 0.9$ | $41.1 \pm 0.4$ | $39.3 \pm 1.4$ | $17.4 \pm 0.7$ | $38.7 \pm 3.5$ | $41.1 \pm 0.5$ |

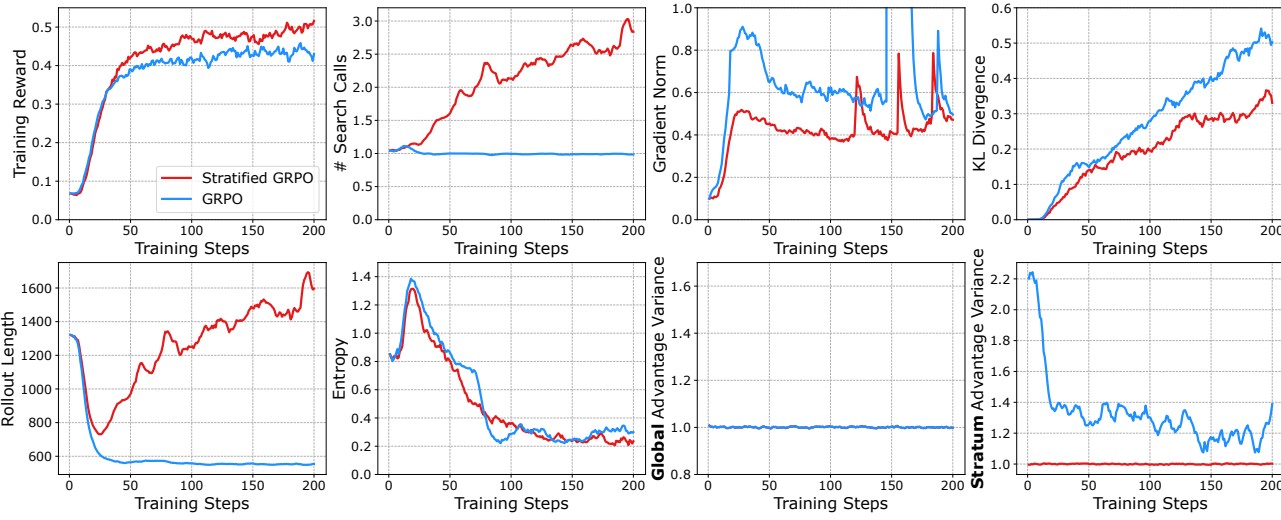

*Figure 2.* Training dynamics comparing Stratified GRPO to GRPO on Qwen 2.5 3B Base, with results averaged across three individual runs. The eight plots show: (1) training reward, (2) search calls per question, (3) gradient norms, (4) KL divergence over training, (5) rollout length, (6) entropy, (7) global advantage variance, and (8) in-stratum advantage variance.

## D.2. Alternative Stratification Rule

To further examine whether the proposed stratified normalization is tied to a single hand-designed stratifier, we test an alternative trajectory-length-based stratification rule on Qwen-2.5-3B-Base. Specifically, we split trajectories into two strata

according to whether their length is larger than 1000 tokens. This rule is also structure-aware, but is less directly tied to external evidence acquisition than tool-call count.

Table 9 shows that length-based stratification improves over GRPO, but remains weaker than tool-call-count stratification. This suggests two points. First, the benefit of SAN is not restricted to one hand-picked proxy, since another reasonable structural variable can also help. Second, the quality of the stratification rule matters: in the search-agent setting studied in this paper, tool-call count is a more direct and effective proxy for trajectory heterogeneity.

*Table 9.* Alternative stratification based on trajectory length on Qwen-2.5-3B-Base. Length stratification uses two strata split by whether the trajectory length is larger than 1000 tokens.

| Method | NQ | TQA | PopQA | HP | 2Wiki | Mus | Bam | Avg. |
|---|---|---|---|---|---|---|---|---|
| GRPO | 45.2 | 61.2 | 43.8 | 32.6 | 29.7 | 7.8 | 12.9 | 33.3 |
| Length Stratification | 45.8 | 61.3 | 41.9 | 40.2 | 33.8 | 14.5 | 31.5 | 38.4 |
| Search-Count Stratification | 45.9 | 61.4 | 43.0 | 40.8 | 39.9 | 17.7 | 42.7 | 41.6 |

### D.3. Additional Sensitivity to the Blending Coefficient

In the main paper, we report the sensitivity of the blending coefficient $\alpha$ on Qwen-2.5-3B-Base. To further examine whether the method relies on delicate tuning of $\alpha$, we additionally sweep $\alpha$ on Qwen-2.5-3B-Instruct. As shown in Table 10, SAN-only already substantially improves over GRPO, and the blended estimator remains strong across a reasonable range of $\alpha$. This supports the interpretation that $\alpha$ acts as a finite-sample stabilizer rather than a fragile task-specific knob.

*Table 10.* Sensitivity of the blending coefficient $\alpha$ on Qwen-2.5-3B-Instruct.

| $\alpha$ | NQ | TQA | PopQA | HP | 2Wiki | Mus | Bam | Avg. |
|---|---|---|---|---|---|---|---|---|
| 0.0 (GRPO) | 33.4 | 52.9 | 36.7 | 26.5 | 27.4 | 6.4 | 21.0 | 29.2 |
| 0.7 | 43.2 | 57.4 | 39.9 | 36.9 | 32.7 | 16.1 | 33.1 | 37.0 |
| 0.8 | 44.5 | 60.9 | 44.3 | 41.0 | 37.3 | 16.9 | 38.7 | 40.5 |
| 0.9 | 42.5 | 59.3 | 42.5 | 40.1 | 37.4 | 16.9 | 36.3 | 39.3 |
| 1.0 (SAN-only) | 42.5 | 60.1 | 44.2 | 39.4 | 41.0 | 16.0 | 36.3 | 39.9 |

## E. Limitations

This work focuses on GRPO-style training for search and tool-use agents whose trajectories exhibit structural heterogeneity. Our formulation of SAN applies to any fixed stratum mapping, and our experiments instantiate it with tool call count as a simple and effective stratifier. This design provides a controlled way to study structural normalization, while leaving richer sources of trajectory heterogeneity to future work. In addition, our theoretical analysis characterizes SAN at the population level, whereas the practical blended estimator bridges this analysis with finite-sample training by improving stability when realized strata are sparse or imbalanced.

## F. Reproducibility Statement

To support the reproducibility of Stratified GRPO, we have discussed implementation details and key resources in the main text and appendix in detail. All algorithmic details are described in Section 3 and Algorithm 1. Full training and evaluation details are specified in Section 5 and Appendix C. All benchmarks (NQ, TriviaQA, PopQA, HotpotQA, 2WikiMultiHopQA, MuSiQue, Bamboogle, and GAIA) and base models (Qwen-2.5-3B Base/Instruct and Qwen3-8B) are publicly available. We report ablations, sensitivity analyses, and multi-seed stability experiments in Section 5.3 and Appendix D. Our code will be available at https://github.com/JIA-Lab-research/Stratified-GRPO

