# OpenReview forum: "Stratified GRPO: Handling Structural Heterogeneity in Reinforcement Learning of LLM Search Agents"
_ICML.cc/2026/Conference — ICML 2026 regular_

### Official Review · Reviewer_1FEg · 2026-03-12

**Soundness:** 3
**Presentation:** 4
**Significance:** 3
**Originality:** 3
**Overall Recommendation:** 5
**Confidence:** 4

**Summary:**

This paper addresses cross-stratum bias in RL for LLM search agents, where structural heterogeneity can distort credit assignment when using a global baseline. The authors propose Stratified GRPO with Stratified Advantage Normalization to compute advantages locally within homogeneous strata. Their theoretical analysis shows that SAN removes cross-stratum bias. A blending mechanism is introduced for finite-sample stability. Experiments on the QA and GAIA benchmarks show consistent gains over baselines.

**Compliance With Llm Reviewing Policy:**

Affirmed.

**Final Justification:**

The authors consider a significant issue and investigate the concept of structural heterogeneity in RL for LLM agents, with solid empirical gains and a clear framework. The rebuttal partially addresses my main concerns, especially the lack of quantitative analysis of Theorem 3.6 and the limited justification of the stratification choice. Overall, my evaluation remains unchanged.

**Key Questions For Authors:**

1. In Theorem 3.6, the weighting by $1/(\sigma_k + \epsilon)$ may favor low-variance strata. Does this empirically suppress exploration in high-variance strata?
2. Why is the search count the most appropriate choice of structural heterogeneity, rather than alternatives such as tool type or placement?
3. Could the authors quantify how often small strata occur in practice and how much they affect learning?

**Limitations:**

This paper does not adequately discuss the limitations and potential negative societal impact. They should expand limitations across different agent tasks where trajectory heterogeneity has richer structural variations.

**Strengths And Weaknesses:**

# Strengths:
* It clearly identifies and mathematically formalizes cross-stratum bias, and shows why global baselines fail in heterogeneous action spaces.
* SAN with Blended Advantage is simple and easy to implement.
* Demonstrates consistent gains over baselines on multi-hop QA and GAIA benchmarks.
# Weaknesses:
* The method only stratifies by search count, which may not capture broader structural heterogeneity.
* The claim that similar issues affect PPO-like methods is not sufficiently supported, given that PPO in search-r1 achieves a competitive performance against GRPO.

---

> ### Author Rebuttal · Authors · 2026-03-31
>
> We thank the reviewer for the thoughtful and constructive feedback. Below, we address the main concerns.
>
> **W1: The method only stratifies by search count, which may not capture broader structural heterogeneity**
>
> We agree that search count does not exhaust all possible sources of structural heterogeneity. Our claim is therefore not that the current paper fully captures every relevant structural factor. Our intended claim is narrower: SAN addresses heterogeneous rollouts captured by a fixed stratum mapping $S=s(\tau,x)$, and tool-call count is a concrete instantiation used in this paper for search agents because it is a simple, stable, and directly observable first-order proxy in this setting. It reflects whether, and to what extent, a rollout acquires external evidence.
>
> Importantly, the gain does not come from partitioning alone. In Table 4, random stratification is nearly identical to GRPO, whereas structure-aware stratification yields a large improvement. We also tested a preliminary length-based stratification on Qwen-2.5-3B-Base (**Q1** of **Reviewer x1k9**); it improves over GRPO, but remains weaker than search-count stratification (38.4 vs. 41.6 avg.). This suggests that the framework is not restricted to a single hand-picked rule, while also confirming that stratifier quality matters. We will revise the paper to make this scope explicit and add to the limitations that richer heterogeneity sources—such as tool type, tool placement, retrieval quality, and multi-tool interactions—remain important future directions not yet fully modeled in the current study.
>
> **W2: PPO-like claim is not sufficiently supported**
>
> We thank the reviewer for pointing this out. Our theory is specific to global-normalization methods such as GRPO; it does not establish an equivalent theorem for PPO. The PPO-related wording in the current draft is too broad, and we will narrow it accordingly. Many thanks!
>
> **Q1: In Theorem 3.6, does the weighting $1/(\sigma_k+\epsilon)$ suppress exploration in high-variance strata?**
>
> Thank you for this important point. We agree that Theorem 3.6 should be interpreted carefully. Our intended claim is that SAN does not give equal weight to all strata. Rather, Theorem 3.6 is a population-level characterization showing that SAN targets a weighted sum of within-stratum gradients after removing the cross-stratum offset induced by global normalization. It supports peer-consistent credit assignment within strata; it is not a guarantee of uniform treatment across strata.
>
> Empirically, we do not observe suppression of harder regimes. Instead, Stratified GRPO learns substantially more multi-step search behavior than GRPO (Figure 1), showing its large gains on multi-hop QA, and improves strongly on GAIA Level 3. This is the opposite of what we would expect if high-variance / complex strata were being suppressed. We will revise the discussion around Theorem 3.6 to make this interpretation more precise and avoid stronger claims than our evidence supports.
>
> **Q2: Why search count rather than tool type or placement?**
>
> Our claim is not that search count is universally optimal. We chose it because, in the search-agent setting studied here, it is a simple first-order proxy that is naturally discrete and easy to group. Finer criteria, such as tool placement, are plausible and interesting, but it is hard to design robust rules to group trajectories. We therefore view search count as a practical and effective choice in the current setting.
>
> **Q3: How much do small strata affect learning?**
>
> We agree that the practically important question is how sparse or imbalanced realized strata affect learning. This is exactly why Section 4 introduces the blended estimator. The current ablations already provide direct evidence about the impact on learning: SAN-only improves over GRPO, while the blended version performs better than SAN-only (Table 3), indicating that finite-sample noise from sparse / imbalanced realized strata is real and that blending helps mitigate it. The $\alpha$-sweep further shows a broad, robust region rather than a sharply tuned optimum (Table 5), suggesting that this effect is meaningful but not catastrophic. We will revise the paper to state this point more explicitly.
>
> **Q4: Limitations and broader impacts are under-discussed**
>
> We agree and will substantially strengthen this part. In the revision, we will explicitly discuss: (1) richer heterogeneity sources beyond search count; (2) the role of blending when realized strata are sparse; and (3) the broader societal impact.

---

> > ### Author Rebuttal · Reviewer_1FEg · 2026-04-03
> >
> > The rebuttal addresses some of my concerns. I appreciate the clarification that the method is designed for a fixed stratum mapping and that the PPO-related claim will be stated more narrowly. The comparison between random and structured stratification, as well as the SAN and blending ablations, is also useful.
> >
> > However, my main concerns are not fully resolved. The discussion of the potential under-weighting of high-variance strata in Theorem 3.6 remains qualitative and does not provide a quantitative assessment. The decision to focus on search count rather than other structural factors is still only partly justified. Overall, the response is helpful, but it is not sufficient to change my evaluation, so I keep my original score unchanged.

---

> > > ### Author Response · Authors · 2026-04-08
> > >
> > > We thank the reviewer for the helpful follow-up.
> > >
> > > **W1: The discussion of the potential under-weighting of high-variance strata in Theorem 3.6 remains qualitative and does not provide a quantitative assessment**
> > >
> > > We agree that Theorem 3.6 should be interpreted carefully. In our paper, Theorem 3.6 is stated for a fixed prompt $x$ and a fixed stratum mapping $S=s(\tau,x)$, so it is a prompt-conditional population characterization. Likewise, SAN is computed per prompt and per stratum, rather than across different prompts. Therefore, the factor $1/(\sigma_k+\epsilon)$ in Theorem 3.6 should be interpreted as a within-prompt normalization-induced scaling, not as a direct statement about practical mixed-prompt minibatch training dynamics.
> > >
> > > To make this distinction more concrete, we examined, at each of 200 training steps on Qwen-2.5-3B-Base under practical mixed-prompt minibatch training with Stratified GRPO, which search-count stratum had the largest empirical within-stratum reward variance. The highest-variance stratum is not fixed: among search-count strata $\{0,1,2,3,4\}$, it appears in 0.5\%, 24.5\%, 22.0\%, 27.5\%, and 25.5\% of the analyzed steps, respectively. Thus, no single stratum persistently dominates batch variance throughout training. Intuitively, it is also unsurprising that the 0-search stratum rarely has the largest variance, since trajectories without search tend to be behaviorally simpler and do not branch on retrieved evidence; however, this is not central to our argument here.
> > >
> > > We would like to keep the claim here narrow. This diagnostic does not by itself establish the absence of any practical imbalance. Rather, it supports a more limited point: Theorem 3.6 should not be interpreted as imposing a persistent penalty on one fixed stratum throughout training.
> > >
> > > As indirect empirical context, Fig. 1 (right) shows that Stratified GRPO learns progressively more multi-step search behavior over training. We view this as inconsistent with a strong, persistent practical suppression of complex search behavior, but not as a direct test of per-stratum update weighting.
> > >
> > > We would therefore like to keep the claim here narrow. Theorem 3.6 is not a full practical exploration guarantee, and we will revise the paper to make this scope explicit and avoid over-interpreting it in that way. Our intended point is that the theorem should not be read as predicting a persistent penalty on any single stratum in practice, especially since the identity of the “highest-variance stratum” is itself dynamic during training.
> > >
> > >
> > > **W2: The decision to focus on search count rather than other structural factors is still only partly justified**
> > >
> > > We agree that this point should be justified more concretely. Our claim is not that search count is universally optimal, but rather that it is a particularly practical and effective choice in the search-agent setting studied here.
> > >
> > > Beyond the random-stratification control in the paper—which is nearly identical to GRPO and therefore suggests that the gain does not come from partitioning alone—we also tested a length-based stratification on Qwen-2.5-3B-Base (**Q1** of **Reviewer x1k9**). Using a two-strata split ($>1000$ tokens vs. $\leq 1000$ tokens), it improves over GRPO, but remains clearly weaker than search-count-based stratification (38.4 vs. 41.6 average). This provides useful evidence for two points:
> > > (1) the SAN framework is not tied to a single hand-picked rule, since another reasonable structural proxy can also help;
> > > (2) stratifier quality matters, and in our setting search count is empirically the stronger choice. The random-stratification result in the paper is consistent with this interpretation: arbitrary grouping does not help, while structure-aware grouping does.
> > >
> > > One plausible reason is that, for search agents, search count is a directly observable, discrete, and semantically direct proxy for how much external evidence a trajectory conditions on, whereas length-based stratification requires an additional thresholding/bucketing choice that is less canonical and may mix more heterogeneous behaviors. In this sense, we do not claim that search count is the only valid stratifier, but it is a cleaner and more robust first-order proxy in the present setting.
> > >
> > > We will revise the paper to reflect this narrower scope more explicitly, and note that richer alternatives remain important directions for future work.

---

### Official Review · Reviewer_ej8H · 2026-03-13

**Soundness:** 3
**Presentation:** 3
**Significance:** 3
**Originality:** 2
**Overall Recommendation:** 4
**Confidence:** 4

**Summary:**

This paper studies reinforcement learning for LLM search agents and argues that their trajectories are structurally heterogeneous, that is, trajectories with different numbers or patterns of search calls may have very different reward distributions and should not be compared using a single global baseline. The paper formalizes this issue as cross-stratum bias and proposes Stratified GRPO, the key component of which is Stratified Advantage Normalization (SAN), i.e., trajectories are partitioned into strata based on structural properties (e.g., search count), and advantages are normalized within each stratum rather than globally. The paper further proposes a blended estimator that interpolates between SAN and global normalization for improved finite-sample stability. The authors provide theoretical claims about eliminating cross-stratum bias, obtaining conditional zero-mean/unit-variance signals within each stratum, and preserving certain global properties. Empirically, the method is evaluated on seven single-hop and multi-hop QA benchmarks using Qwen-2.5-3B Base/Instruct search agents, and it reports sizable gains over GRPO and several other baselines, especially on multi-hop tasks.

**Compliance With Llm Reviewing Policy:**

Affirmed.

**Final Justification:**

From a theoretical perspective, I think that the stratified idea in this paper is  **very direct and not particularly novel**, especially since it has already been well established in the statistics literature. However, the paper is overall **very complete**, and I am therefore inclined toward a **weak accept**.

**Key Questions For Authors:**

>**Questions**
1. How sensitive is the method to the choice of stratification instead of trajectories with different length?
2. What objective is the blended practical estimator optimizing, and how should readers interpret the theoretical guarantees for the final algorithm?
3. How does performance change when strata are small or highly imbalanced?

**Limitations:**

The paper should discuss limitations more explicitly. In particular, it should acknowledge that:
+ the method may be highly dependent on the hand-designed stratification rule.
+ gains are currently shown in a relatively specific search-QA training setup with small models and limited training scale;
+ the practical method relies on blending partly because pure stratification can become noisy in small strata; and
+ the broader societal impact is probably limited but still includes the usual concerns around more effective automated search agents, such as misuse for scaled information extraction or the amplification of retrieval errors. A more explicit limitations section would improve the paper.

**Strengths And Weaknesses:**

>**Strengths**

The paper identifies a real modeling issue in RL for search agents, that is, trajectories with different search behaviors can indeed have different reward statistics, so using one global baseline may lead to poor credit assignment. The decomposition in Proposition 1 or Theorem 1 is intuitive and mathematically clean, and the practical method is simple enough that the empirical gains are easy to attribute to the proposed modification rather than a large engineering stack. The experiments cover both base and instruct variants and include both single-hop and multi-hop QA, where the largest gains appear on harder multi-hop tasks. The paper also includes an ablation between GRPO, SAN-only, and the blended version, which is useful for understanding where the improvements come from. I do think the problem setting is important. RL for tool-using/search agents is increasingly relevant, and handling heterogeneity in rollouts is practically meaningful. A simple modification that improves stability and exploration in search-heavy settings could be useful to practitioners, particularly if it transfers beyond this exact benchmark suite.

> **Weaknesses**

My main concern is novelty. At a high level, the method amounts to grouping trajectories into strata and normalizing advantages within each group, with an additional convex blend for small-sample stability. This is a very **direct** application of classical variance-reduction or stratification intuition[1,2] to GRPO. The paper packages this as “cross-stratum bias,” but much of the underlying phenomenon may be viewed as a fairly expected consequence of mixing heterogeneous trajectories under one baseline. Therefore, while the paper is technically reasonable, the algorithmic idea feels incremental unless the authors can better justify why this insight is non-obvious and substantially different from standard grouped/conditional baselines.

[1] Adaptive Strategy for Stratified Monte Carlo Sampling, Journal of Machine Learning Research(JMLR), 2015

[2]Stratified sampling meets machine learning, International conference on machine learning(ICML), 2016

---

> ### Author Rebuttal · Authors · 2026-03-31
>
> We thank the reviewer for the thoughtful and constructive feedback. Below, we address the main concerns.
>
> **W1: Novelty: relation to classical stratification / grouped baselines**
>
> Thanks for bringing [1, 2]. We agree that our method is related in spirit to classical stratification, and we do not claim to have invented stratification as a general idea. Our intended novelty claim is narrower and setting-specific.
>
> In classical stratified sampling, strata are fixed **before** sampling and the algorithm allocates budget across them. In our setting, the structural variable of interest (e.g., number of tool calls) is only known **after** a trajectory is generated. Thus, we cannot pre-allocate samples by stratum in the classical sense. Our method instead performs an ex-post, structure-aware normalization on realized trajectories for credit assignment.
>
> Our contribution is not simply that grouping may help. We identify and formalize a specific failure mode of GRPO-style global normalization for LLM search agents: same-prompt rollouts can differ substantially in structure and reward statistics, and a single global baseline can distort within-stratum credit assignment. Empirically, random stratification performs nearly the same as GRPO, while structure-aware stratification yields substantial gains, suggesting that the gains are not explained by arbitrary grouping alone.
>
> **Q1: Sensitivity to the choice of stratification**
>
> We agree that our claim is not that arbitrary stratification rules should work equally well. Stratified normalization is expected to help when the chosen variable captures a **meaningful source of structural heterogeneity**.
>
> Our current evidence is consistent with this interpretation. In Table 4, random stratification performs nearly the same as GRPO, suggesting that the gains do not come from partitioning trajectories per se, but from grouping according to a variable aligned with the actual source of heterogeneity. In the current search-agent setting, we use tool-call count because it is simple, directly observable, naturally discrete, and closely tied to how much external evidence a trajectory conditions on.
>
> As additional **preliminary** evidence (**Q1** of **Reviewer x1k9**), we also tested a length-based stratification and observed improvement over GRPO, although it remained weaker than tool-call-count stratification. We view this as suggestive rather than definitive: it indicates that the effect is not tied to a single hand-picked proxy, while also confirming that stratifier quality matters. At the same time, trajectory length is less canonical here because it requires an additional thresholding choice to form strata.
>
> **Q2: What objective does the blended estimator optimize?**
>
> The blended estimator does not change the underlying RL objective. The optimization target remains the expected reward objective in Eq. (1). The practical method changes only the advantage estimator used in the policy-gradient update: $A_{\text{blend}}(\tau)=\alpha A_{\text{SAN}}(\tau)+(1-\alpha)A_{\text{GN}}(\tau).$
>
> We agree that the exact results in Section 3 characterize SAN itself, not the blended estimator. The blended variant is introduced in Section 4 as a finite-sample stabilization for cases where realized strata can be small. Therefore, the final algorithm should be interpreted as using a stabilized surrogate gradient estimator motivated by SAN’s analysis, rather than as inheriting SAN’s exact guarantees verbatim.
>
> **Q3: Small or highly imbalanced strata**
>
> We agree that this is a practical concern. In our setting, stratum occupancies are not fixed in advance: they are jointly determined by the current policy and the prompt distribution, and therefore evolve during training. For this reason, we do not claim a closed-form characterization in terms of a single imbalance parameter.
>
> Our current evidence is therefore indirect and empirical rather than a dedicated imbalance-controlled study. First, Section 4 introduces blending precisely because realized strata can be small or imbalanced in finite-sample training, which can make pure SAN noisy. Second, our ablations are consistent with this interpretation: SAN-only improves over GRPO, while the blended version improves further. Third, the $\alpha$-sweep shows a fairly broad robust region (Table 5), suggesting that the practical method is not overly sensitive once some amount of global stabilization is included.
>
> **Q4: Limitations and broader impacts**
>
> We agree that the current draft should discuss limitations and broader impacts more explicitly. In the revision, we will add a dedicated discussion addressing the raised four points: (i) dependence on the stratification rule, (ii) the current evaluation scale being relatively modest, (iii) the role of blending when realized strata are small, and (iv) broader societal impact, including possible misuse and the amplification of retrieval errors.

---

> > ### Author Rebuttal · Reviewer_ej8H · 2026-04-03
> >
> > The authors have addressed my concerns well in the rebuttal, and I appreciate their clarifications. Therefore, I would like to maintain my original score.

---

### Official Review · Reviewer_x1k9 · 2026-03-13

**Soundness:** 3
**Presentation:** 3
**Significance:** 3
**Originality:** 2
**Overall Recommendation:** 4
**Confidence:** 3

**Summary:**

This paper studies reinforcement learning for LLM search agents, where sampled trajectories can differ substantially in structure due to different numbers of tool calls. The authors argue that standard GRPO uses a single global normalization across such heterogeneous trajectories, which leads to “cross-stratum bias” and harms credit assignment and exploration. To address this, they propose Stratified GRPO, whose main idea is to partition rollouts into strata based on structural properties, primarily the number of tool calls, and then normalize advantages within each stratum via Stratified Advantage Normalization (SAN). They further introduce a blended estimator that interpolates between SAN and the original global normalization for finite-sample stability. The paper also provides a theoretical decomposition of the standard normalized advantage into a stratified term plus a cross-stratum offset, along with conditional moment analysis and a population-level gradient interpretation. Empirically, the method is evaluated on seven factual QA benchmarks and the GAIA benchmark, where it improves over GRPO in the reported settings.

**Compliance With Llm Reviewing Policy:**

Affirmed.

**Final Justification:**

The authors have addressed my concerns well in the rebuttal, and I appreciate their clarifications. Therefore, I would like to raise my score to 4.

**Key Questions For Authors:**

1. Your method depends critically on the chosen stratification variable, and all experiments use tool-call count. How sensitive is performance to alternative structure definitions, such as search placement, query diversity, trajectory length, or retrieved-document properties? A stronger answer here would help establish whether the method is broadly useful or mainly tuned to one specific heterogeneity pattern.

**Limitations:**

See weakness.

**Strengths And Weaknesses:**

**Strengths**

1. The authors consider an important problem for RL. The proposed method showing a stratum-specific offset in global normalization makes the intuition clear and easy to follow.

2. The proposed design is lightweight and easy to implement on top of GRPO. The paper also includes a pragmatic blended variant for finite-sample regimes, which is sensible given that some strata may be small.

3. The paper evaluates both factual QA and a more agentic benchmark (GAIA), and includes ablations on SAN alone, random stratification, and the blending coefficient. These experiments help support that the reported gains are tied to the structure-aware grouping rather than arbitrary partitioning.

**Weaknesses**

1. The method is a rather direct extension of GRPO: replace one global normalization with per-stratum normalization, then blend with the original estimator. While the paper wraps this with theoretical analysis, the core recipe itself feels incremental relative to standard variance-reduction and grouping ideas. I am not yet convinced this rises to the level of a strong conference contribution on its own.

2. In all experiments, stratification is based on the number of tool calls. The authors investigate the concept in this one concrete form, but the paper does not show that the strategy/metric generalizes to other forms of trajectory heterogeneity, other tools, or other agent settings beyond the WebQA & GAIA.

3. The paper should compare with more advanced RL baselines like DAPO to demonstrate its effectiveness because there exist many RL improvement after the relese of all the mentioned baselines.

---

> ### Author Rebuttal · Authors · 2026-03-31
>
> We thank the reviewer for the thoughtful and constructive feedback. Below, we address the main concerns.
>
> **W1: The method may appear incremental relative to GRPO**
>
> Thank you for raising this point. We agree that the algorithmic change itself is intentionally simple, and we do **not** claim that Stratified GRPO is a fundamentally new RL framework. Our contribution is instead to identify and characterize a specific failure mode that arises in RL for **structurally heterogeneous search-agent rollouts**.
>
> The key point is not merely “using groups,” but that in this setting the standard globally normalized advantage can be decomposed into a within-stratum term plus a **stratum-dependent offset**. This offset induces a non-zero conditional mean across strata, which distorts local credit assignment. SAN removes this offset by normalizing among structurally comparable trajectories, and the blended version improves finite-sample stability. In this sense, the contribution is a **diagnosis, characterization, and correction** of a previously unaddressed issue in LLM search-agent RL, rather than additional algorithmic complexity for its own sake.
>
> We will revise the paper to make this framing more explicit and more modest.
>
> **W2: Scope and generalization beyond the current search-agent setting**
>
> We agree that the empirical scope should be stated carefully. The paper is primarily about **search agents**, and we do **not** intend to claim empirical validation for all RL training scenarios for LLMs.
>
> Within this scope, the current experiments already cover two distinct search-agent regimes:
> (1) retrieval-based factual QA over Wikipedia, and
> (2) GAIA-style deep-research tasks involving open-web search and long-horizon reasoning.
>
> We will revise the paper to present these results as evidence **within the search-agent domain**, and avoid wording that could be read as claiming broader generality than is currently supported.
>
> As additional supporting evidence in our response to **W1** of **Reviewer vvtJ**, we also observed that Stratified GRPO with the same tool-call-count stratification achieves qualitative improvement over GRPO for VLM with the zoom-in tool; however, we view this only as supporting evidence consistent with the mechanism, not as a basis for expanding the paper’s main empirical claim beyond search agents.
>
> **W3: Comparison with stronger RL baselines like DAPO**
>
> We thank the reviewer for this suggestion. We add a **DAPO** comparison on Qwen-2.5-3B-Base under the same factual-QA setup. DAPO improves GRPO from **33.3 to 35.5** average, while **DAPO + Ours further improves to 43.8**. These results suggest that our method is **compatible with** stronger RL baselines rather than being redundant with them.
>
> | Method | NQ | TriviaQA | PopQA | HotpotQA | 2Wiki | Musique | Bamboogle | Avg |
> |---|---:|---:|---:|---:|---:|---:|---:|---:|
> | GRPO | 45.2 | 61.2 | 43.8 | 32.6 | 29.7 | 7.8| 12.9 | 33.3 |
> | Ours | 45.9 | 61.4 | 43.0 | 40.8 | 39.9 | 17.7 | 42.7 | 41.6 |
> | DAPO | 46.9 | 62.3 | 45.9 | 34.0 | 31.2 | 8.5 | 19.4 | 35.5 |
> | DAPO w/ Ours | 48.1 | 63.6 | 47.2 | 44.2 | 42.7 | 19.4 | 41.1 | 43.8 |
>
> We will add this result in the revision.
>
> **Q1: Alternative stratification rules**
>
> We agree this is an important question. Our claim is **not** that arbitrary stratification rules should work equally well. Rather, stratified normalization is expected to help only when the chosen variable captures a **meaningful source of structural heterogeneity**—i.e., trajectories that differ systematically in behavior and reward statistics and therefore should not be normalized against one another. If a candidate variable does **not** induce such heterogeneity, we would not expect a meaningful gain over global normalization. This is consistent with our random-stratification ablation, which performs nearly the same as GRPO.
>
> A second practical consideration is that not all candidate variables are easy to discretize into stable rollout groups. Since our current formulation assumes a fixed and stable stratum assignment, we focus on tool-call count: in the search-agent setting studied here, it is simple, observable, and easy to group.
>
> As a preliminary sensitivity check, we also tested an alternative based on trajectory length on Qwen-2.5-3B-Base, using a two-strata split (**>1000 tokens** vs. **≤1000 tokens**):
>
> | Method | NQ | TriviaQA | PopQA | HotpotQA | 2Wiki | Musique | Bamboogle | Avg |
> |---|---:|---:|---:|---:|---:|---:|---:|---:|
> | GRPO | 45.2 | 61.2 | 43.8 | 32.6 | 29.7 | 7.8| 12.9 | 33.3 |
> | Ours w/ length stratification | 45.8 | 61.3 | 41.9 | 40.2 | 33.8 | 14.5 | 31.5 | 38.4 |
> | Ours w/ search count stratification | 45.9 | 61.4 | 43.0 | 40.8 | 39.9 | 17.7 | 42.7 | 41.6 |
>
> It improves over GRPO, but is still weaker than tool-call-count stratification, suggesting that the effect is not unique to one hand-picked proxy while also confirming that **stratifier quality matters**.

---

> > ### Author Rebuttal · Reviewer_x1k9 · 2026-04-07
> >
> > The authors have addressed my concerns well in the rebuttal, and I appreciate their clarifications. Therefore, I would like to raise my score.

---

### Official Review · Reviewer_vvtJ · 2026-03-13

**Soundness:** 2
**Presentation:** 3
**Significance:** 2
**Originality:** 3
**Overall Recommendation:** 4
**Confidence:** 3

**Summary:**

The paper introduces Stratified GRPO, a modification of the GRPO reinforcement learning algorithm designed for training LLM-based search agents. The authors observe that trajectories generated by such agents often differ structurally. Standard GRPO computes advantages using a single global baseline over all trajectories, which can lead to biased comparisons between structurally different trajectories. To address this issue, the proposed method partitions trajectories into strata based on structural properties, specifically the number of tool calls. Advantage normalization is then performed within each stratum, ensuring that trajectories are compared only with structurally similar ones. Experimental results on multiple question-answering and agent benchmarks demonstrate that Stratified GRPO consistently outperforms standard GRPO in overall performance.

**Compliance With Llm Reviewing Policy:**

Affirmed.

**Final Justification:**

The response clarifications address most of my concerns. I raise my score to 4.
One remaining point is generality. Although the method is mainly designed for search agent RL, this application constraint can be a limitation for this work.

**Key Questions For Authors:**

See weakness.

**Limitations:**

Yes

**Strengths And Weaknesses:**

Strength:

1. The proposed Stratified Advantage Normalization (SAN) is conceptually straightforward: partition trajectories by structure and compute advantages locally, making it easy to implement on top of existing GRPO pipelines.

2. The writing is clean and easy to read.

3. The evaluation on search tasks shows strong performance compared to other baselines on search tasks.

Weakness:

1. Stratification rule. Strata are defined solely by the number of tool calls, which is task-specific and heuristic. It is unclear whether this rule generalizes to other settings, especially tasks with many tool interactions or tasks without tool use (e.g., math tasks).


2. Hyperparameter sensitivity. The blending coefficient α is introduced to combine stratified and global advantages. Although an ablation study is provided, it is limited to one model and setup, and it remains unclear whether a single α works robustly across tasks or requires tuning.


3. Limited task diversity. Experiments focus primarily on search-based QA and agent tasks. It is unclear whether the proposed approach generalizes to other RL training scenarios for LLMs beyond search agents. A potential issue is that such strata strategy can only work on the evaluated search tasks.

---

> ### Author Rebuttal · Authors · 2026-03-31
>
> We thank the reviewer for the careful reading and constructive feedback. Below we address the main concerns.
>
> **W1: Stratification rule: “number of tool calls” may be heuristic and task-specific**
>
> We agree that we should explain this design choice more clearly. Our claim is not that the number of tool calls is the only valid stratification rule. Rather, SAN itself is defined for an arbitrary fixed stratum mapping $S=s(\tau,x)$, while tool-call count is the concrete instantiation used in this paper for search agents. We use it because, in the search-agent setting, it is the simplest observable proxy for structural heterogeneity: it reflects whether, and to what extent, a rollout chooses to acquire external evidence. This choice substantially affects the trajectory structure, the retrieved information, and the resulting reward distribution, which is exactly the type of heterogeneity SAN is designed to handle.
>
> We also emphasize that the gains do not come from partitioning alone. In Table 4 of the paper, random stratification performs nearly identically to GRPO, whereas structure-aware stratification yields a large improvement. This suggests that the benefit comes from capturing a meaningful heterogeneity driver in search trajectories, rather than from introducing an arbitrary partition.
>
> To further examine whether the method is overly tied to the search agent setup, we additionally ran an experiment on Qwen3-VL-2B-Instruct in a recent **Thinking with Images** setting [1, 2], where a VLM is equipped with a zoom-in tool for inspecting local regions of high-resolution images. We used the same tool-call-count stratification, and observed consistent gains over GRPO:
>
> | Model|V*|HR-Bench 4K|HR-Bench 8K|MME-RealWorld|Avg|
> |-|-:|-:|-:|-:|-:|
> |Qwen3-VL-2B-Instruct|72.3|67.8|62.8 |41.6|61.1|
> |+GRPO|78.0|68.1|63.3 |46.0|63.9|
> |+Stratified GRPO|79.6|70.6|65.4|49.1|66.2|
>
> We view this as additional supporting evidence that the same stratification rule can remain effective in another task, while keeping the main focus of the paper on search agents. In the revision, we will make this distinction more explicit, better motivate why tool-call count is the natural first choice in the current search-agent setting, and include the experimental details for the Thinking with Images setting.
>
> **W2: Sensitivity of the blending coefficient $\alpha$**
>
> We thank the reviewer for highlighting the importance of hyperparameter robustness. Our intent was not to introduce $\alpha$ as a fragile task-specific knob, but as a finite-sample stabilizer on top of SAN.
>
> The current ablation on Qwen-2.5-3B-Base (Table 5 of the paper) already shows two encouraging facts:
> (1) SAN-only ($\alpha=1$) already improves over GRPO; and
> (2) performance remains strong over a reasonably broad range ($\alpha \in [0.4,0.8]$), rather than peaking sharply at a single value.
>
> To further address this concern, we additionally ablated $\alpha$ on Qwen-2.5-3B-Instruct:
>
> | Value of $\alpha$|NQ|TriviaQA|PopQA|HotpotQA|2Wiki|Musique|Bamboogle|Avg|
> |-|-:|-:|-:|-:|-:|-:|-:|-:|
> |0.0 (GRPO)|33.4|52.9|36.7|26.5|27.4|6.4|21.0|29.2|
> |0.7|43.2|57.4|39.9|36.9|32.7|16.1|33.1|37.0|
> |0.8|44.5|60.9|44.3|41.0|37.3|16.9|38.7|40.5|
> |0.9|42.5|59.3|42.5|40.1|37.4|16.9|36.3|39.3|
> |1.0 (SAN-only)|42.5|60.1|44.2|39.4|41.0|16.0|36.3|39.9|
>
> These results show that SAN-only already improves over GRPO, and that performance stays strong over a reasonable range. The evidence suggests that the method is not highly sensitive to $\alpha$ and does not appear to rely on delicate tuning to outperform GRPO.
>
> **W3: Limited task diversity and concern that the method may only work on the evaluated search tasks**
>
> We agree that the empirical scope should be stated carefully. Our paper primarily studies search agents, rather than all RL training scenarios for LLMs.
>
> Within this scope, the current experiments already cover substantially different settings:
> - retrieval-based factual QA, and
> - a deep-research agent on GAIA with open-web search using Google.
>
> Across both settings, Stratified GRPO consistently improves over standard GRPO. Therefore, even within the search-agent scope, the current evidence is already broad.
>
> In addition, the extra Thinking with Images result above suggests that the improvement is not confined to one text-only setup. We view this result as supporting evidence consistent with the underlying mechanism—namely, that comparing structurally different rollouts against one global baseline can hurt credit assignment—rather than as a basis for expanding the main empirical claim of the paper beyond search agents.
>
> In the revision, we will make the scope more explicit and avoid overstating breadth beyond what is currently supported by the experiments.
>
> [1] DeepEyes: Incentivizing "Thinking with Images" via Reinforcement Learning, ICLR 2026.
>
> [2] Pixel Reasoner: Incentivizing Pixel-Space Reasoning with Curiosity-Driven Reinforcement Learning, NeurIPS 2025.

---

> > ### Author Rebuttal · Reviewer_vvtJ · 2026-04-06
> >
> > Thank you for the response. The clarifications address most of my concerns. I raise my score to 4.
> > One remaining point is generality. Although the method is mainly designed for search agent RL, this application constraint can be a limitation for this work.

---

### Decision · Program_Chairs · 2026-04-30

**Decision:**

Accept (regular)

**Comment:**

The reviewers agree on three things: First, the paper tackles a real problem: mixing structurally different trajectories under one global baseline genuinely hurts credit assignment, and the paper formalizes this cleanly. Second, the fix is refreshingly simple, just normalizing advantages within structurally similar groups rather than globally, making it easy to drop into existing GRPO pipelines. Third, the experiments are convincing, showing consistent gains across multiple QA benchmarks and the GAIA agent tasks, with ablations that confirm the improvements come from meaningful stratification rather than arbitrary grouping. The remaining concerns around the theoretical interpretation of Theorem 3.6 and generality beyond search agents are acknowledged limitations but not dealbreakers, as confirmed by the reviewers in the post-rebuttal discussion. I would thus recommend acceptance.